# A pause in the weakening of the Atlantic meridional overturning circulation since the early 2010s

**Sang-Ki Lee** [1] ✉, **Dongmin Kim** [1,2], **Fabian A. Gomez** [1,3], **Hosmay Lopez** [1], **Denis L. Volkov** [1,2], **Shenfu Dong** [1], **Rick Lumpkin** [1] & **Stephen Yeager** [4]

The current state-of-the-art climate models when combined together suggest that the anthropogenic weakening of the Atlantic Meridional Overturning Circulation (AMOC) has already begun since the mid-1980s. However, continuous direct observational records during the past two decades have shown remarkable resilience of the AMOC. To shed light on this apparent contradiction, here we attempt to attribute the interdecadal variation of the historical AMOC to the anthropogenic and natural signals, by analyzing multiple climate and surface-forced ocean model simulations together with direct observational data. Our analysis suggests that an extensive weakening of the AMOC occurred in the 2000s, as evident from the surface-forced ocean model simulations, and was primarily driven by anthropogenic forcing and possibly augmented by natural variability. However, since the early 2010s, the natural component of the AMOC has greatly strengthened due to the development of a strong positive North Atlantic Oscillation. The enhanced natural AMOC signal in turn acted to oppose the anthropogenic weakening signal, leading to a near stalling of the AMOC weakening. Further analysis suggests that the tug-of-war between the natural and anthropogenic signals will likely continue in the next several years.

The Atlantic Meridional Overturning Circulation (AMOC) is the Atlantic component of the global ocean conveyor belt, a large-scale ocean circulation system that carries heat, salt, carbon, and other biogeochemical elements along its paths[1,2]. Thus, the AMOC is a crucial component of the global heat, salt, nutrients and carbon balances affecting the associated regional climate, sea-level, and marine ecosystems[3–9]. For instance, the AMOC-related carbon sinking in the high-latitude North Atlantic accounts for a significant portion of the anthropogenic carbon inventory of the global ocean[10–12]; thus, the amount of anthropogenic carbon sequestered by the ocean is closely tied to the strength of the AMOC[13–16].

The climate models participating in the Coupled Model Intercomparison Project Phase 6 (CMIP6), when combined together, display a slow increase in the AMOC during the historical period until around the mid-1980s primarily due to increased anthropogenic aerosol loading, and a decline afterward due to increased $CO_2$ and reduced aerosol loading[17–19]. The decline in CMIP6 models from 1985 to 2014 amounts to about −2.3 Sv[17]. In reference to the observed mean value of 16.7 Sv at 26.5°N (between January 2005 and December 2021)[20–22], it represents a 14% decrease per 30 years, which is within the range of the projected rate of weakening during 2015 − 2100 (i.e., 6 - 8 Sv or 13 - 17% per 30 years)[23]. Hence, the historical AMOC evolution during the past several decades may provide invaluable insights into the projected future weakening of the AMOC.

The observed AMOC at 26.5°N was relatively strong in 2005 (19.2 Sv) but decreased rapidly in 2009 (14.6 Sv) and 2010 (14.9 Sv).

[1]NOAA Atlantic Oceanographic and Meteorological Laboratory, Miami, FL, USA. [2]Cooperative Institute for Marine and Atmospheric Studies, University of Miami, Miami, FL, USA. [3]Northern Gulf Institute, Mississippi State University, Mississippi State, MS, USA. [4]National Center for Atmospheric Research, Boulder, CO, USA. ✉e-mail: Sang-Ki.Lee@noaa.gov

After 2010, however, it gradually recovered[20-22] until it peaked in 2018 (18.0 Sv) and then weakened again until 2021 (15.3 Sv), showing virtually no trend during 2011-21 (−0.1 Sv per decade). Although an overall decreasing trend is found for the entire observational period (−1.1 Sv per decade during 2005-21), the linear trend is sensitive to the relatively strong AMOC values in the initial years. For instance, removing the first two years, the linear trend during 2007-21 becomes only −0.2 Sv per decade. Consistently, the time-mean AMOC at 26.5°N in the first 10 years (2005-14) is reduced by only −0.2 Sv in the latter 7 years (2015-21), thus displaying a strong resilience at the interdecadal time scale. A recent update of the submarine cable measurement record of the Florida Current transport, a critical component of the AMOC, further reduces the weakening trend of the annual mean AMOC during 2005-21 by about 40% (i.e., from −1.1 Sv per decade to −0.6 Sv per decade)[24].

Although there is no long-term continuous observational record of the AMOC prior to April 2004, several studies based on a hydrographic data-constrained model, a high-resolution ocean reanalysis, and multiple ocean models forced by reanalysis surface flux fields suggested an interdecadal swing of the historical AMOC similar to that in CMIP6 models[25-30]. However, those studies concluded that the historical AMOC swing was predominantly modulated by natural interdecadal climate variability associated with the North Atlantic Oscillation (NAO), although several proxy-based studies suggested a longer-term, thus presumably anthropogenic, weakening of the AMOC since the mid- or late-19th century[31-34]. As such, currently there is no strong consensus regarding the relative roles of natural and anthropogenic drivers in the past AMOC changes[35]. For that reason, it is quite uncertain when the projected weakening of the AMOC will emerge, or if it has already emerged, and can be detected.

Here, we use a suite of climate and ocean model simulations along with direct observational data to attribute the interdecadal variation of the historical AMOC since the mid-1950s to the externally forced signal (i.e., largely anthropogenic forcings but also includes solar and volcanic forcings) and natural signal. Specifically, the RAPID/Meridional Overturning Circulation and Heatflux Array/Western Boundary Time Series (RAPID) moored array[22] and 10 surface-forced ocean model simulations participating in the Ocean Model Intercomparison Project Phase-2 (OMIP2[27], Supplementary Table 1) are analyzed to derive the historical AMOC; two sets of large ensemble climate model simulations, based on the Community Earth System Model version 2 (CESM2)[36], and the Seamless System for Prediction and Earth System Research (SPEAR)[37], and 20 CMIP6 model simulations (Supplementary Tables 2 and 3) are analyzed to derive the externally forced signal; and the difference between the historical and externally forced AMOC components is analyzed to estimate the natural signal (Methods). After separating the natural signal from the externally forced one, we further examine if the time evolution of the natural AMOC is a physically consistent response to its main driver - interdecadal NAO variability[38-45]. Finally, we discuss the potential decadal predictability of the natural AMOC component at 26.5°N and the expected range of the AMOC at 26.5°N during the next several years.

## Results

### Interdecadal variations of the AMOC components at 26.5°N

Figure 1a shows the decadally averaged time series of the AMOC anomalies at 26.5°N from the decade centered in 1960 (i.e., 1955-64) to the decade centered in 2050 (i.e., 2045-54), derived from CESM2 (green), OMIP2 (blue) and RAPID (magenta). To allow a better visual comparison of the three interdecadal AMOC time series, the time-averaged RAPID AMOC transport value of 16.7 Sv is removed from each time series (Methods). The AMOC values derived from OMIP2 and RAPID are referred to as the total component, and those from the ensemble mean of CESM2 as the externally forced component. The differences between the two (i.e., total - externally forced) are referred to as the residual

component (orange bars in Fig. 1a), which would represent natural variability provided OMIP2 and CESM2 yield realistic levels of the total and externally forced AMOC components, respectively. It should be noted that the relative strengths of the three AMOC components are sensitive to the AMOC adjustments (i.e., bias corrections) in OMIP2 and CESM2 (Methods); thus, our analysis is for the most part based on the rate of interdecadal change in the AMOC (Fig. 1b) since it is not affected by the adjustments in OMIP2 and CESM2. Hereafter, the decade centered in a particular year is simply denoted as the centered year (e.g., the period of 1985−94 is denoted as 1990).

As shown in Fig. 1a, the total and externally forced AMOC components show a similar interdecadal swing with an increase from 1960 to 1990 and a decrease from 2000 to 2020. This suggests that the interdecadal variation in the AMOC from 1960 to 2020 represents a response to the external forcing. However, the total AMOC peaks in 2000, whereas the externally forced AMOC component peaks in 1980-90. Additionally, the internal variability in the CESM2 ensembles (i.e., error bars in Fig. 1a, Methods) is much larger than the externally forced signals; thus, the interdecadal AMOC swing from 1960 to 2020 could also be of natural origin[46-48]. Indeed, the total increase from 1960 to 1990 is much greater than the externally forced increase during that period, whereas the total decrease from 2000 to 2020 is much weaker than the externally forced decrease during that period (Fig. 1a, b). Thus, the residual component is strongly negative prior to 1990 and positive afterward (Fig. 1a).

More specifically, the residual component contributes +1.1 Sv to the total increase between 1960 and 1990. Thus, the total increase in the AMOC from 1960 to 1990 (+2.2 Sv) can be partitioned into a +1.1 Sv increase (50%) in the externally forced component and a +1.1 Sv increase (50%) in the residual component (Fig. 1b). It appears that the external forcing plays a major role in increasing the AMOC from 1960 to 1980 (Fig. 1b). However, a steep increase in the residual component from 1980 to 1990 (+1.6 Sv) is almost completely responsible for the AMOC increase during that period (Fig. 1b). After reaching its peak in 1980−90, the externally forced component starts decreasing. From 1990 to 2000, it decreases by −0.6 Sv. However, the total AMOC continues to increase moderately by +0.4 Sv due to a large increase in the residual component (+1.0 Sv). Thus, the external forcing is not the primary driver of the AMOC increase from 1980 to 2000 in agreement with previous studies[49,50].

As indicated in Fig. 1a and b, the decrease in the externally forced AMOC component from 2000 to 2020 (i.e., due to increased $CO_2$ and reduced aerosol loading) is −2.6 Sv. This is mitigated by a +1.0 Sv increase in the residual component, yielding a −1.6 Sv decrease in the total. Although this is somewhat small (i.e., a 10% of the total), it is important to note that the external forcing is the sole driver of the AMOC slowdown from 2000 to 2020. This result is in disagreement with the conventional argument that the reduction of the AMOC after its peak in 2000 is largely due to the concurrent weakening of the NAO[25-30]. In contrast to that argument, the residual component remains positive and continues to increase overall after 1990, despite a small decrease from 2000 to 2010, and thus largely mitigates the externally forced weakening of the AMOC after 1990 (Fig. 1a, b). The mitigating effect of the residual component becomes quite strong during the recent period from 2010 to 2020 (+1.1 Sv); thus, the externally forced weakening of the AMOC (−1.3 Sv) is nearly compensated for during that period (Fig. 1b).

The interdecadal variation of the externally forced AMOC component derived from CESM2 is largely consistent with that derived from CMIP6 and SPEAR (Fig. 2). However, several differences are noteworthy. For instance, the rate of weakening in the externally forced component (after 1990) is generally smaller in CMIP6 compared to that of CESM2 and SPEAR although the inter-model spread is very large among the 20 CMIP6 models used in this study (i.e., error bars in Fig. 2a, Methods). In particular, the rate of interdecadal

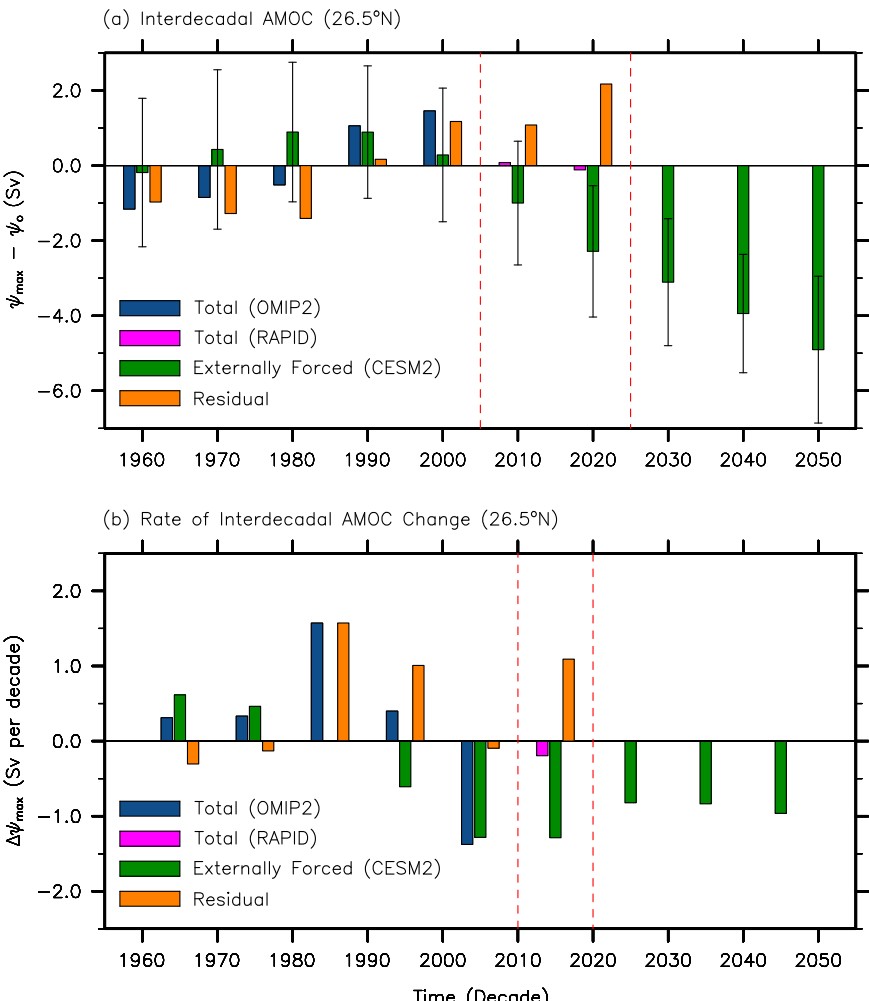

**Fig. 1 | Interdecadal time series of the AMOC and its rate of change at 26.5°N.**
**a** Decade-averaged time series of the Atlantic Meridional Overturning Circulation (AMOC) anomalies at 26.5°N ($\psi_{max} - \psi_o$) from the decade centered in 1960 to the decade centered in 2050, derived from OMIP2 (blue), CESM2 (green) and RAPID (magenta). The time-averaged RAPID AMOC transport value of 16.7 Sv ($\psi_o$) (Methods) is removed from each time series. The AMOC values derived from OMIP2 and RAPID are referred to as the total component, those from the ensemble mean of CESM2 the externally forced component, and the differences between them (i.e., total - externally forced) the residual component (orange). The error bars in (**a**) indicate the 95% limits of the adjusted CESM2 ensemble spread ("Methods"), meaning that about 95% of the adjusted 80 ensembles reside within the error bars. **b** Same as (**a**) except that the rate of interdecadal AMOC change is shown. Red dashed lines in (**a**) and (**b**) indicate the period during which the total AMOC values are derived from RAPID only. The units are $Sv$ ($10^6\ m^3sec^{-1}$) for the AMOC, and $Sv$ per decade for the rate of AMOC changes.

weakening in the externally forced component from 2000 to 2010 is smallest in CMIP6 (−0.8 Sv), slightly larger in SPEAR (−0.9 Sv) and largest in CESM2 (−1.3 Sv). Accordingly, the residual AMOC component decreases from 2000 to 2010 by −0.6 Sv if CMIP6 is used to represent the externally forced component, −0.5 Sv if SPEAR is used, and only −0.1 Sv if CESM2 is used. Thus, the residual component plays an active role in the weakening of the AMOC from 2000 to 2010 (−1.4 Sv) if either CMIP6 or SPEAR is used to represent the externally forced component. See Supplementary Notes 1–2 and Supplementary Figs. 1–5 for more discussion on the sensitivity in attributing interdecadal AMOC variations to model resolution in OMIP2 and CMIP6 models.

In summary, our analysis suggests that the increase in the AMOC at 26.5°N from 1980 to 2000 is mainly driven by an increase in the residual component while the external forcing suppresses the increase in the AMOC from 1990 to 2000. In contrast, the slowdown of the AMOC at 26.5°N from 2000 to 2010 is primarily driven by the external forcing and possibly augmented by the residual component. From 2010 to 2020, however, a large increase in the residual component nearly compensates for the externally forced weakening of the AMOC.

In the next section, we examine if the interdecadal variation of the residual AMOC component can be ratified as a physically consistent response to its main driver - interdecadal NAO variability[38–45].

### Interdecadal variation of the residual AMOC component
The residual component represents not only natural variability of the AMOC, but also deficiencies in the climate and surface-forced ocean models used involving, for instance, the lack of land-ice melting, and the insufficient model resolution for boundary currents, deep water formation, and mesoscale eddies[51]. Therefore, in this section, we investigate if the residual component reasonably represents natural variability despite potential contaminations arising from deficiencies in the climate and surface-forced ocean models used. If we assume that the residual component reflects natural variability to a reasonable degree, Fig. 1a and b suggest that natural variability keeps the AMOC in a relatively weakened state from 1960 to 1980 and then produces a steep increase from 1980 to 1990 in tandem with the external forcing during 1960-80. Then, it works against the external forcing to mitigate the slowdown of the AMOC from 1990 to 2020. As such, the residual AMOC component at 26.5°N increases greatly from 1980 to 2020

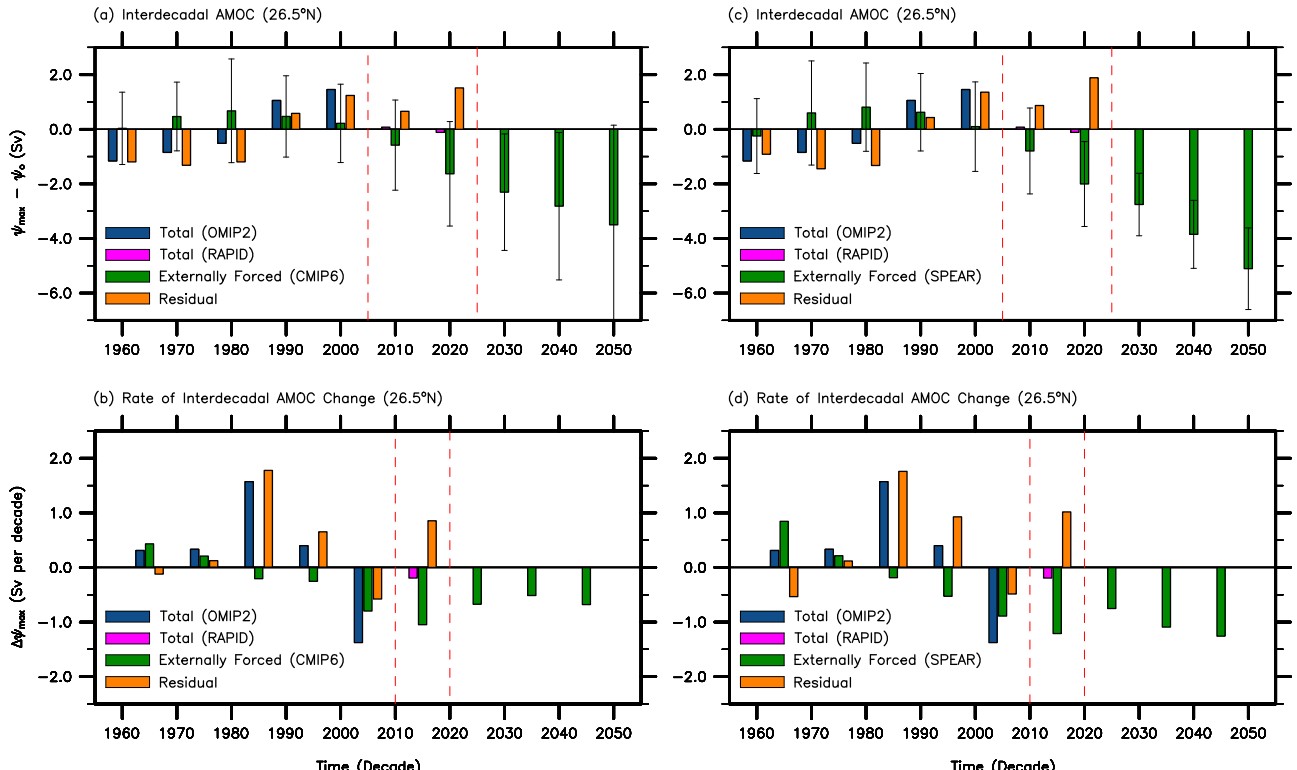

**Fig. 2 | Interdecadal time series of the AMOC and its rate of change at 26.5°N based on CMIP6 and SPEAR. a, b** Same as Fig. 1a, b but the externally forced Atlantic Meridional Overturning Circulation (AMOC) component is derived from CMIP6. **c, d** Same as Fig. 1a, b but the externally forced AMOC component is derived from SPEAR. The error bars in (**a**) indicate the 95% limits of the CMIP6 multi-model spread (Methods), meaning that about 95% of the 20 CMIP6 models reside within the error bars. Similarly, the error bars in (**c**) indicate the 95% limits of the SPEAR ensemble spread (Methods), meaning that about 95% of the adjusted 30 SPEAR ensembles reside within the error bars.

(+3.6 Sv). Figure 1b further shows that the increase occurs mainly during three interdecadal periods from 1980 to 1990 (+1.6 Sv), from 1990 to 2000 (+1.0 Sv), and from 2010 to 2020 (+1.1 Sv). As shown in Fig. 2, if either CMIP6 or SPEAR is used to represent the externally forced component, the corresponding transports of the residual AMOC component show similar interdecadal variations.

Given the well-known physical relationship between the NAO-induced surface buoyancy flux, deep convection in the subpolar North Atlantic, and the AMOC[38–45], we examine if the interdecadal increases in the residual AMOC component at 26.5°N during the 1980s, 1990s and 2010s (Figs. 1b, 2b and d) can be explained as a physically consistent response to interdecadal NAO variability. As indicated in Fig. 3a, a strong negative December-May NAO prevailed from 1960 to 1970. After a pause (i.e., a neutral NAO) in 1980, it switched to a strong positive NAO in 1990, and then weakened to a neutral NAO in 2000 and 2010. This was followed by the development of another strong positive NAO in 2020 (Methods, Supplementary Note 3, and Supplementary Fig. 6).

The development of a strong positive NAO in 1990 from a neutral NAO in 1980 aligns well with the large increase in the residual AMOC component from 1980 to 1990. Similarly, the development of another strong positive NAO in 2020 from a neutral NAO in 2010 (i.e., 2005-14) matches well with the large increase in the residual AMOC component from 2010 to 2020. However, no such direct relationship between the NAO and residual AMOC component is found in other interdecadal periods. For instance, the strong negative NAO in 1960 and 1970 turned into a neutral NAO in 1980, but the residual AMOC component remains strongly negative in 1980. Additionally, the strong positive NAO in 1990 turned into a neutral NAO phase in 2000. However, the residual AMOC component persists and even increases from 1990 to 2000.

Thus, it appears that when a strong NAO develops and continues for a decade, it provides sufficient momentum and duration for the

AMOC to spin up during that decade. Furthermore, if the strong NAO recedes into a neutral NAO in the next decade, the NAO-driven residual AMOC component tends to persist and even intensify throughout the neutral phase that follows. These results are by and large consistent with the NAO leading the AMOC at 26.5°N by 3 – 15 years in the majority of climate and surface-forced ocean models[41,43,44]. Thus, at interdecadal time scales, the residual AMOC component at 26.5°N may be roughly regarded as a time-integrated response to the NAO (see Fig. 3a) as proposed in previous studies[52–55]. More specifically, the NAO-driven surface buoyancy flux is integrated into ocean density changes in the subpolar North Atlantic producing a meridional ocean density gradient anomaly between the subpolar and subtropical North Atlantic Ocean. The NAO-driven surface buoyancy flux continues to drive the anomalous density gradient until the NAO turns into a neutral phase. Thus, the anomalous ocean density gradient increases until a strong NAO turns into a neutral phase, and then remains at that amplitude while the neutral NAO condition continues. The anomalous meridional density gradient in turn should drive the residual AMOC component via dynamic ocean adjustment processes with a time delay. Thus, the residual AMOC component at 26.5°N can be described as a delayed ocean dynamic adjustment to the time-integrated NAO (hereafter, a time-integrated & delayed response to the NAO). However, if the neutral NAO continues for an extended period, the anomalous density gradient should be eventually dissipated by the anomalous meridional heat and salt transports by the residual AMOC component; thus, the residual AMOC component should also be dissipated, but it may take much longer than a decade for that to happen.

To validate the above rationale, we use the December-May NAO and the residual AMOC component derived from the 80-member CESM2 ensembles to carry out a composite analysis. First, the NAO index time series is computed and averaged for each decade from 1850

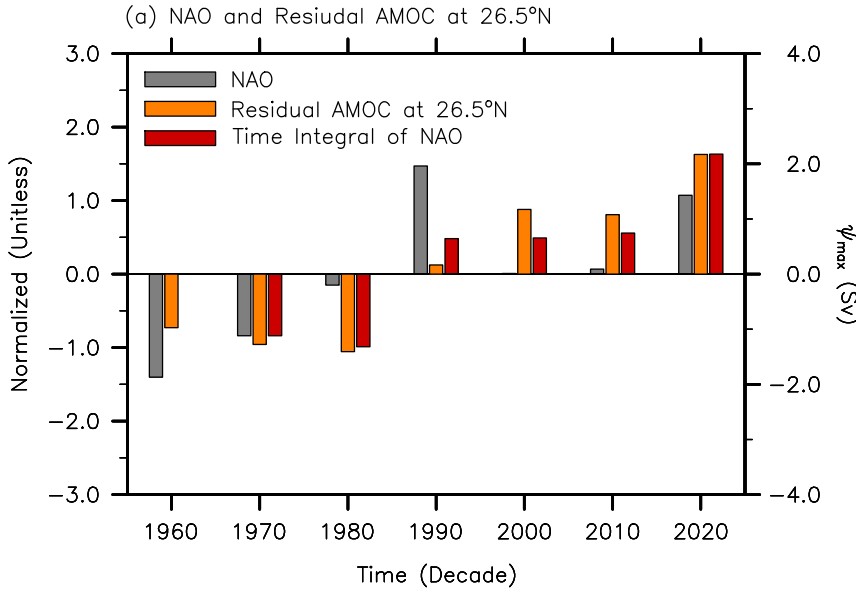

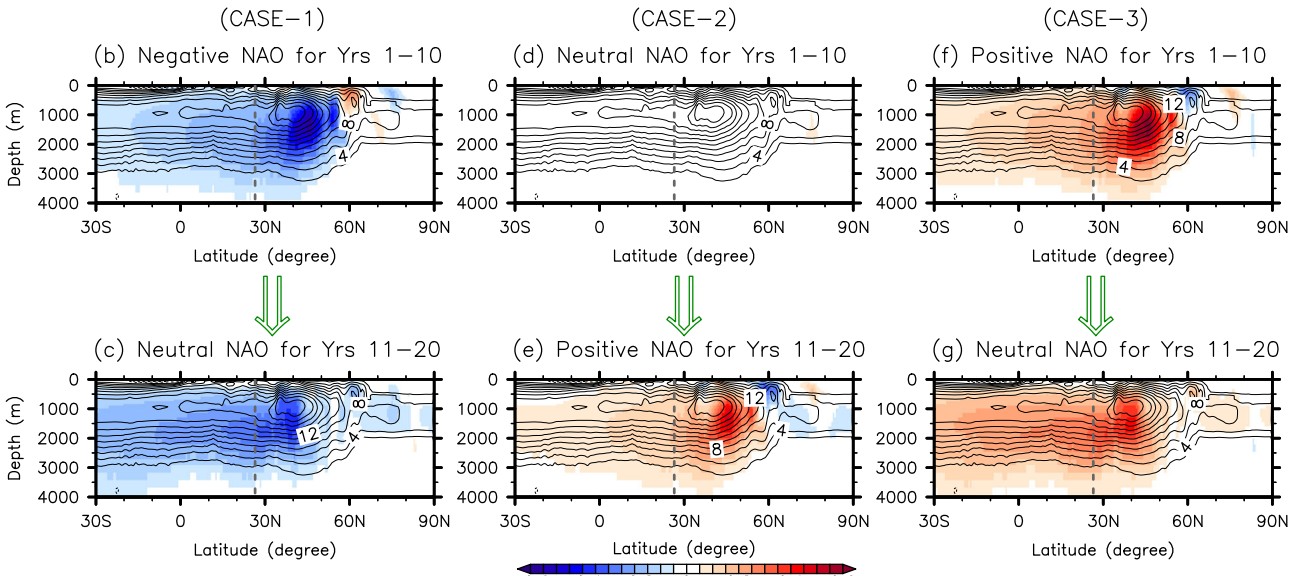

**Fig. 3 | Relationship between the NAO and residual AMOC component.**
**a** Decade-averaged time series of the December-May North Atlantic Oscillation (NAO, gray), the time-integrated NAO (red), and the residual Atlantic Meridional Overturning Circulation (AMOC) component at 26.5°N (orange). The NAO time series is derived from ERA5. The time integrated NAO is computed by accumulating the time series of the December-May NAO from 1970. **b**–**g** The responses of the residual AMOC component to (case-1) a negative NAO for the first 10 years (**b**) followed by a neutral NAO for the latter 10 years (**c**), to (case-2) a neutral NAO for the first 10 years (**d**) followed by a positive NAO for the latter 10 years (**e**), and to (case-3) a positive NAO for the first 10 years (**f**) followed by a neutral NAO for the latter 10 years (**g**). The gray dashed lines in (**b**–**g**) indicate the latitude of 26.5°N. The composite analysis is carried out based on CESM2. The NAO and time-integrated NAO indices are normalized by the corresponding standard deviation, and thus are unitless. The units for the residual AMOC component are $Sv$ ($10^6$ $m^3 sec^{-1}$).

to 2020 for the entire 80 ensembles to derive a total of 1360 decadal NAO index samples (Methods). These samples are then sorted to identify positive NAO (upper-terciles), neutral NAO (mid-terciles) and negative NAO cases (lower-terciles). The first composite is under a negative NAO during the first decade followed by a neutral NAO during the subsequent decade (case-1). The second composite is under a neutral NAO during the first decade followed by a positive NAO in the subsequent decade (case-2). The third composite is under a positive NAO phase during the first decade followed by a neutral NAO during the subsequent decade (case-3).

As shown in Fig. 3b–g, if a negative NAO develops and persists for a decade, the residual AMOC component decreases during that decade (case-1). Similarly, if a positive NAO develops and continues for a

decade, the residual AMOC component increases during that decade (case-3). If the negative or positive NAO recedes into a neutral NAO during the subsequent decade, the NAO-driven residual AMOC component largely persists in the subtropical North Atlantic throughout the neutral phase that follows, and even intensifies in some latitudes (cases-1 and −3). Lastly, if a neutral NAO persists for a decade followed by a positive NAO for the subsequent decade, the residual AMOC component increases during the latter decade (case-2). Thus, the composite analysis supports the relationship between the NAO and the residual AMOC component at 26.5°N shown in Fig. 3a and thus the rationale that at interdecadal time scale the residual AMOC component at 26.5°N can be regarded as a time-integrated & delayed response to the NAO.

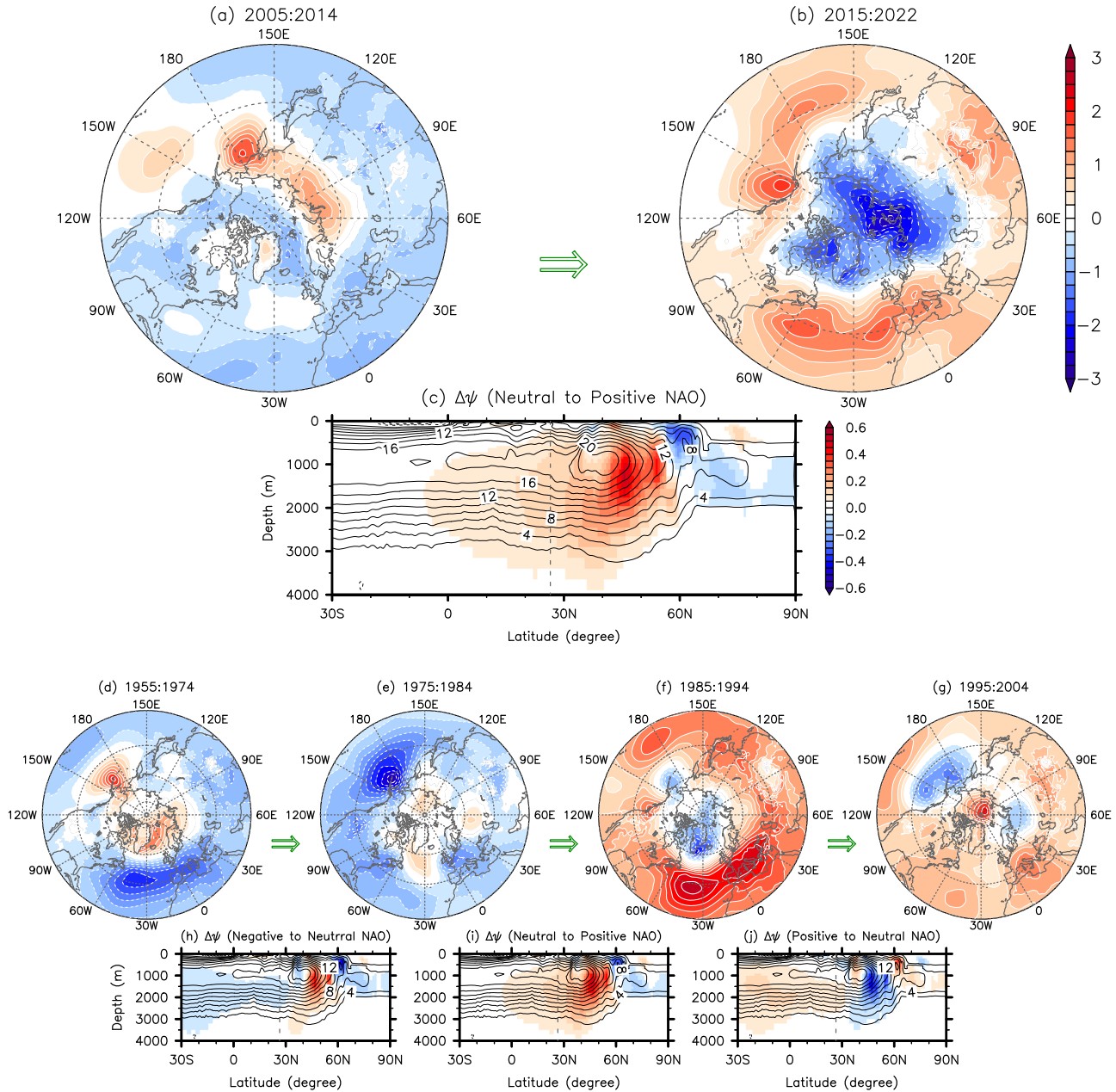

**Fig. 4 | Interdecadal sea level pressure anomalies and the implied rates of change in the residual AMOC component. a, b, d–g** Sea level pressure anomalies in December-May for each decade from 1955-64 to 2015-22 derived from ERA5 and **c, h–j** the implied rates of interdecadal change (i.e., the difference between the first 10 years and the latter 10 years) in the residual Atlantic Meridional Overturning Circulation (AMOC) component (shades) derived from the composite analysis. The contours in (**c, h–j**) indicate the time averaged total AMOC derived from CESM2. The gray dashed lines in (**c, h–j**) indicate the latitude of 26.5°N. The units are *hPa* for sea level pressure anomalies, and *Sv* ($10^6$ $m^3sec^{-1}$) per decade for the rate of interdecadal change in the residual AMOC component.

It is worthwhile to note that the residual AMOC component at 26.5°N decreases from 2000 to 2010 (Figs. 1b, 2b & 2d) although the extent of the decrease depends on the climate model used to represent the externally forced component. This decrease appears to be a dissipation of the elevated residual AMOC component in response to the prolonged near-neutral NAO condition during 2000 and 2010 (Fig. 3a).

An important point to note from the case-1 and case-3 is that while the residual AMOC component remains strong in the subtropical region during the neutral NAO period in the latter decade, it tends to dissipate in the subpolar region during that period. This result is in agreement with previous modeling studies[40,43], and suggests that the residual AMOC component in the subpolar region responds much more quickly and directly to the NAO-driven deep convection

anomalies compared to the residual AMOC component in the subtropical region. See Supplementary Fig. 7 for the responses of the residual AMOC component to the nine different possible cases of interdecadal NAO variability.

To summarize, we show in Fig. 4 the sea level pressure anomalies in December-May for each decade from 1960 to 2020 derived from the European Center for Medium-Range Weather Forecasts reanalysis-5 (ERA5)[56] and the implied interdecadal changes (i.e., the difference between the first and subsequent decades) in the residual AMOC component derived from the composite analysis. The development of a strong positive NAO in 1990 from a neutral NAO in 1980 increases the residual AMOC component from 1980 to 1990, and the development of another strong positive NAO in 2020 from a neutral NAO in 2010 (i.e., 2005-14) increases the residual AMOC component from 2010 to

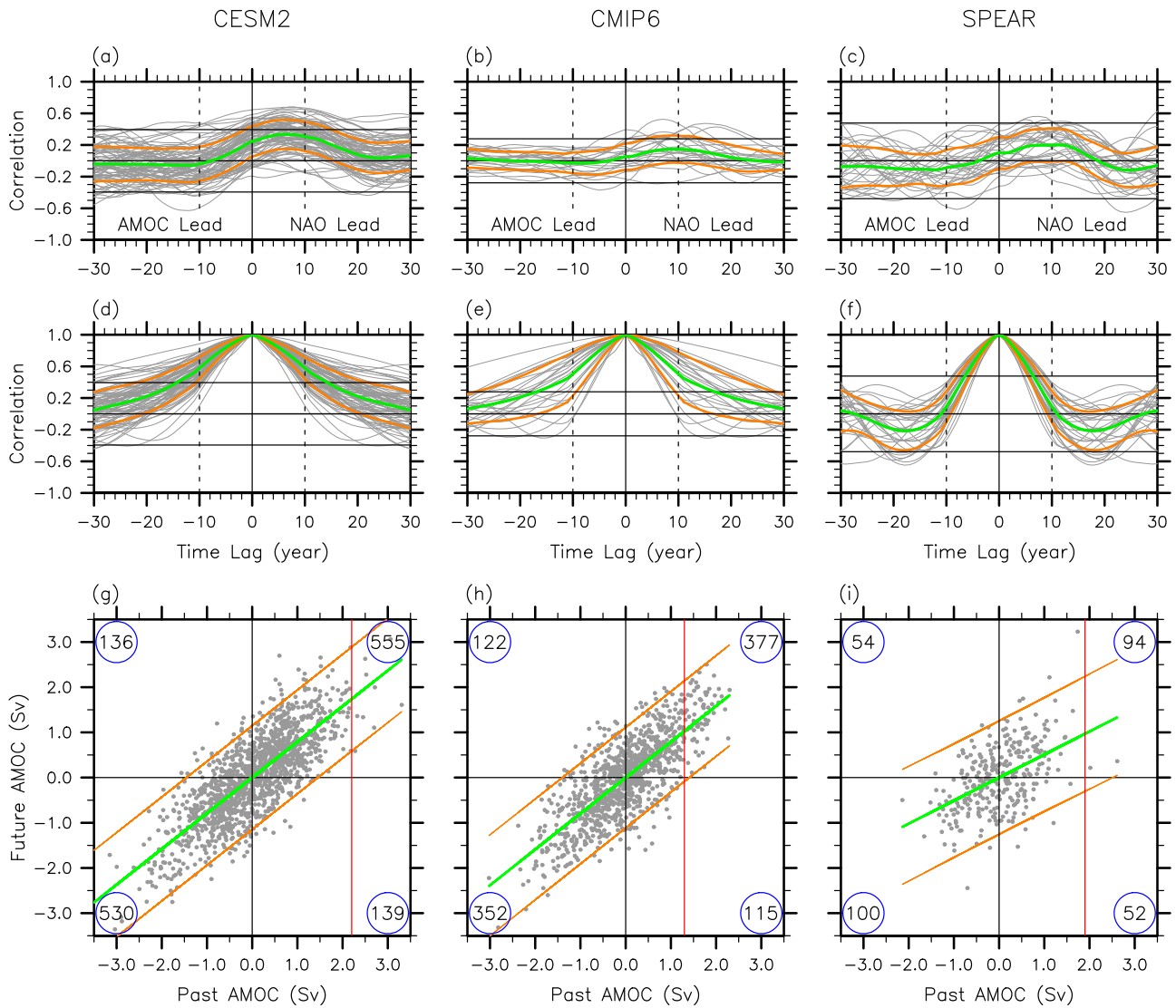

**Fig. 5 | Lead-lag correlations of the NAO and the residual AMOC component at 26.5°N and autocorrelations of the residual AMOC component at 26.5°N.** **a–c** Lead-lag correlations between the North Atlantic Oscillation (NAO) and the residual Atlantic Meridional Overturning Circulation (AMOC) component at 26.5°N, and (**d–f**) the autocorrelations of the residual AMOC component derived from (**a, d**) CESM2, (**b, e**) CMIP6 and (**c, f**) SPEAR. The green lines in (**a–f**) indicate the ensemble-averaged correlation values. The orange lines in (**a–f**) indicate the ensemble spread (i.e., standard deviation). Thick horizontal lines in (**a–f**) indicate the 95% confidence intervals (based on a Student-*t* test) and zero correlations. The degree of freedom for the Student-*t* test is determined by the number of decades since the NAO and AMOC indices are smoothed by performing a 10-year running-

average prior to the correlation analysis to focus on interdecadal time scale. **g–i** The residual AMOC component values at 26.5°N averaged for each decade are plotted against the residual AMOC component values for the corresponding next 1-5 years, derived from (**g**) CESM2, (**h**) CMIP6 and (**i**) SPEAR. The green lines in (**g–i**) indicate the mean slopes of the residual AMOC component between a past decade and the next 1–5 years. The pair of orange lines in (**g–i**) indicate the 95% limits of the regressed residual AMOC component values. The red vertical lines in (**g–i**) indicate the residual AMOC component values at 26.5°N averaged for the period of 2015-21. The numerical values at the four corners of (**g–i**) indicate the number of dots within each of the four quadrants. The units for the residual AMOC component are *Sv* (10⁶ *m³sec⁻¹*).

2020, consistent with case-2. The strong negative NAO in 1960 and 1970 turned into a neutral NAO in 1980, but the residual AMOC component remains negative and slightly decreases further in 1980 in the subtropical North Atlantic, consistent with case-1. Lastly, the strong positive NAO in 1990 turned into a neutral NAO in 2000, but the residual AMOC component remains positive and increases in the subtropical North Atlantic between 1990 and 2000, consistent with case-3.

**Potential decadal predictability of the residual AMOC component**

The externally forced AMOC weakening is projected to continue in the next decade and beyond throughout the rest of the 21st century[23]. Between 2020 and 2030, for instance, the AMOC at 26.5°N is expected

to weaken by −0.8 Sv according to CESM2 (Fig. 1a, b). Therefore, in reference to the observed value of the AMOC during 2015-21 (i.e., 16.6 Sv), the expected mean value of the AMOC at 26.5°N in 2030 is 15.8 Sv assuming that the residual AMOC component in 2020 remains the same in the next decade. A similar value of 15.9 Sv is projected from both CMIP6 and SPEAR. However, based on the composite analysis and previous studies[40,41,43,44], the residual AMOC component at 26.5°N may stay elevated or even further increase during the next decade as a time-integrated & delayed response to the strong positive NAO in 2020. Thus, we explore in this section the potential decadal predictability of the residual AMOC component at 26.5°N.

We first explore the lead-lag correlations between the December-May NAO index and the residual AMOC component at 26.5°N derived from all 80-member CESM2 ensembles. As shown in Fig. 5a, the

correlations tend to be maximized when the NAO leads the residual AMOC component by 0‑15 years. For some ensemble members, the maximum correlations are quite high exceeding the 95% confidence level. However, when all 80 ensemble members are merged together (green line), the maximum correlation drops below the 95% confidence level, in agreement with an earlier study based on the Coupled Model Intercomparison Project Phase 5 (CMIP5) models[44]. Therefore, despite the positive correlation between the NAO and residual AMOC component, the time history of NAO is not a reliable predictor of the residual AMOC component at 26.5°N in CESM2 (see Supplementary Fig. 8). A consistent result is found when either 22 CMIP6 models under the preindustrial scenario (Supplementary Table 3) or 30 ensemble members of SPEAR are used to compute the lead-lag correlations between the December-May NAO index and the residual AMOC component at 26.5°N (Fig. 5b,c). This result is surprising given the critical role of the NAO in the interdecadal AMOC variation in OMIP2[44]. A previous study pointed out that the NAO simulated in fully coupled models tends to display weaker-than-observed interdecadal variations, and thus its influence on interdecadal AMOC variations may be underestimated[42].

We next explore if the history of the residual AMOC component serves as a potential predictor for the future evolution of the residual AMOC component. Figure 5d shows the autocorrelations of the residual AMOC component derived from the CESM2 ensembles. When all 80 ensemble members are merged together (green line), the autocorrelation, which has the zero-crossing at the time lead-lag of around 30-year, is statistically significant above the 95% confidence level at the time lead-lag of 13-year and shorter. Additionally, for the majority of the ensemble members (i.e., 71 out of 80), the autocorrelation values are above the 95% confidence level at the time lead-lag of 10-year, suggesting that the history of the residual AMOC component is a potential source for predicting the future residual AMOC component. This result is not very surprising since the time histories of the NAO and other drivers of the residual AMOC component are implicitly integrated into the residual AMOC component of the past. Additionally, the initialization of the AMOC has been shown to be a useful source of decadal predictability of, if there are any, potential internal oscillations[57]. A similar result is found from the 22 CMIP6 preindustrial runs (Fig. 5e). However, the autocorrelation from the 30-ensemble SPEAR, which has the zero-crossing at the time lead-lag of around 10-year, is significant only for the subsequent 1-6 years (Fig. 5f).

Based on the above result, the values of the residual AMOC component at 26.5°N averaged for each decade from 1850 to 2020 are plotted against the values of the residual AMOC component for the corresponding subsequent 1-5 years (Fig. 5g). For instance, the residual AMOC component averaged for 2005-14 is plotted against the residual AMOC component averaged for 2015-19. As expected, the residual AMOC component during a past decade appears to be useful for predicting the residual AMOC component in the subsequent 1-5 years. The mean slope is about 0.8. This means that if the residual AMOC component value of +2.2 Sv for 2015-21 is used (Fig. 1a), the expected value of the residual AMOC component during 2022-26 is about +1.7 ± 1.1 Sv, yielding a reduction of −0.5 ± 1.1 Sv. According to CMIP6, the residual AMOC component is reduced from +1.3 Sv in 2015-21 to +1.0 ± 1.1 Sv in 2022-25 (Fig. 5h), yielding a reduction of −0.3 ± 1.1 Sv. Similarly, according to SPEAR, the residual AMOC component is reduced from +1.9 Sv to +1.0 ± 1.3 Sv (Fig. 5i), yielding a reduction of −0.9 ± 1.3 Sv. Therefore, we can conclude based on the above analysis that the highly elevated state of the residual AMOC component at 26.5°N in 2020 is unlikely to dissipate away during the next several years. See Supplementary Table 4 for more details. The autocorrelation-based estimates should not be used as an actual decadal prediction tool because there are skillful decadal dynamic forecast systems with realistic initializations of the ocean[58–60]. Nevertheless, the global annual-to-decadal climate synthesis report issued by the World Meteorological Organization[61] predicted that the total AMOC around 30°N in 2023–27 is likely to be near or slightly below the recently observed values, largely in agreement with our conclusion.

## Discussion

In an effort to attribute the interdecadal AMOC evolution during the historical period to the anthropogenic and natural signals, we analyze a suite of climate and surface-forced ocean model simulations along with direct observational data. As summarized in Fig. 6 (see also Supplementary Figs. 9 and 10), the weakening of the AMOC since 2000, which is evident from the surface-forced ocean model simulations, is primarily driven by the external forcing (i.e., increased $CO_2$ and reduced aerosol loading) and is partly compensated by an increase in the natural component. Focusing on the recent period of direct observations, the AMOC remains nearly unchanged from 2010 to 2020 because the externally forced component and the natural component largely compensate for each other. Further analysis indicates that the natural AMOC component increases mainly during three interdecadal periods from 1980 to 1990, from 1990 to 2000, and from 2010 to 2020 in response to the development of strong positive NAO in 1990, its time-integrated & delayed impact in 2000, and the development of another strong positive NAO in 2020, respectively. Lastly, we show that the highly elevated residual AMOC component at 26.5°N in 2020 is unlikely to dissipate away during the next several years.

There remain many outstanding questions that require further investigations. For instance, it is not explored in this study through what processes or pathways the external forcing drives the weakening of the AMOC since 2000. Previous studies point to several such pathways, including reduced ocean-to-air turbulent heat fluxes over the Greenland and Iceland Seas due to rapidly rising air temperatures[62], a potential impact of retreating sea-ice cover along the regional boundary currents in the Greenland, Iceland and Norwegian Seas[63], and a spreading of Arctic freshwater through Fram and David Straits[64,65]. A freshening of the Labrador Sea after the mid-1990s due to enhanced melting of Greenland's ice sheet has been also suggested as another pathway through which external forcing could drive the weakening of the AMOC[66], although this process is missing in the climate and ocean models used in this study. Thus, it is important to further investigate if and to what extent the above-listed pathways of external forcing influence the anthropogenic weakening of the AMOC in climate and ocean models.

As discussed and shown in our analysis, natural interdecadal AMOC variability appears to be predominantly driven by the NAO. However, in the climate models analyzed in this study, only a small portion of the interdecadal AMOC variance can be explained by the NAO (Fig. 5a–c). Although this may be due to deficiencies in those models[42], it is important to note that the NAO is not the only factor contributing to natural interdecadal AMOC variability. It is also possible that unforced internal ocean processes may also influence interdecadal AMOC variability[67–73]. Indeed, a hydrographic-data constrained model reconstruction of the AMOC[30] shows somewhat different interdecadal AMOC evolution from that in OMIP2. Thus, further research is needed to identify the role of climate modes of variability other than the NAO, such as East Atlantic Pattern[74,75], in natural interdecadal AMOC variability, and to better understand unforced internal AMOC variability and its interaction with the NAO-forced interdecadal AMOC variability.

Several key limitations in this study should be discussed. First of all, the main conclusions in this study heavily rely on OMIP2 and the climate models used (i.e., CESM2, CMIP6 and SPEAR), but these models have limitations stemming from the lack of land-ice melting, and the insufficient model resolution for representing boundary currents, deep water formation, and mesoscale eddies. Additionally, OMIP2 and the climate models used simulate somewhat different mean states of

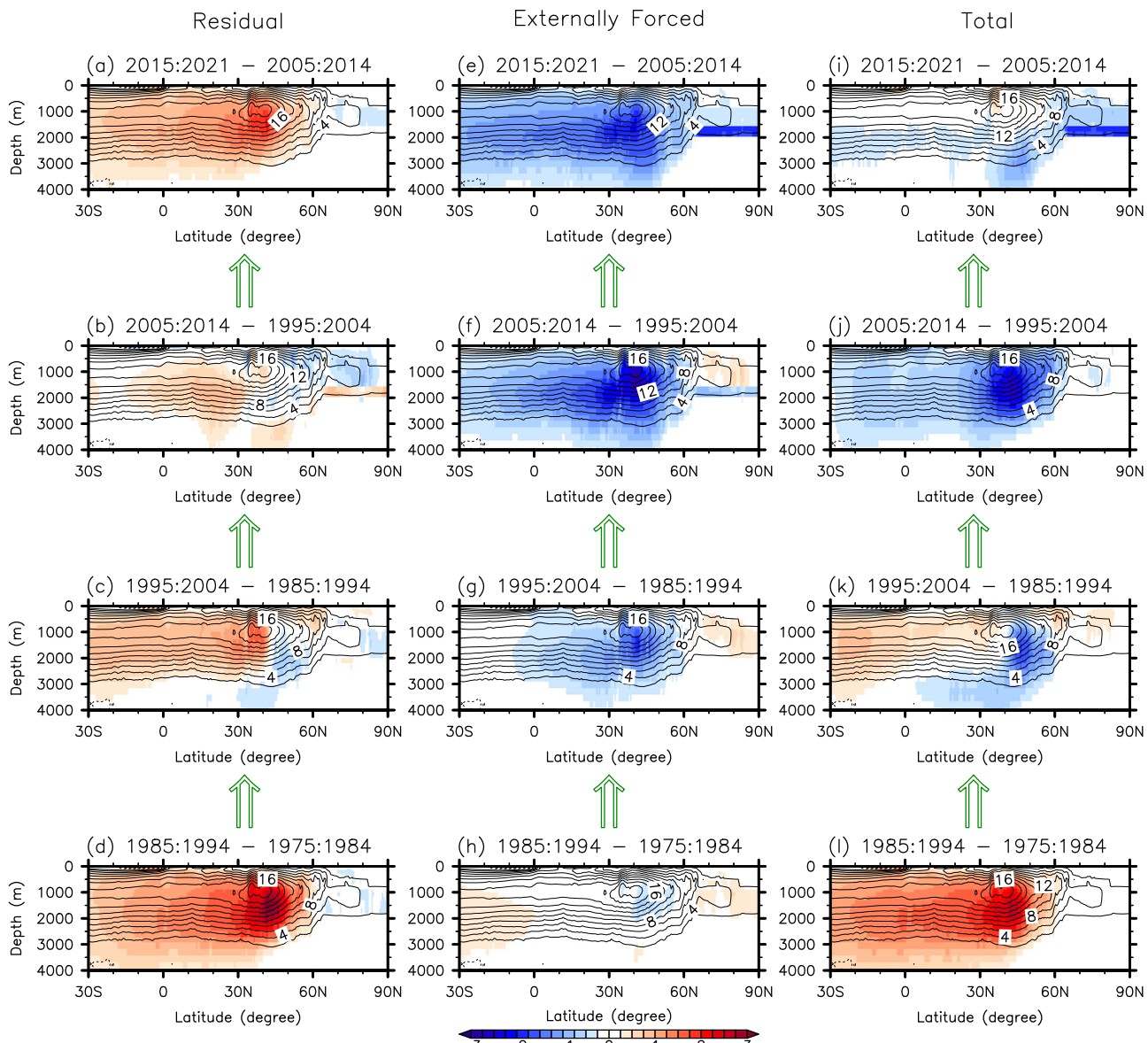

**Fig. 6 | The rates of interdecadal change in the AMOC components.** The rates of interdecadal change (shades) in (**a–d**) the residual, (**e–h**) externally forced, and (**i–l**) total Atlantic Meridional Overturning Circulation (AMOC) components **d**, **h**, **l** from 1975-84 to 1985-94, **c**, **g**, **k** from 1985-94 to 1995-2004, **b**, **f**, **j** from 1995-2004 to 2005-14, and **a**, **e**, **i** from 2005-14 to 2015-21, derived from OMIP2 and the ensemble-mean CESM2. The contours indicate the time averaged total AMOC derived by averaging that of OMIP2 and CESM2. The residual AMOC component for 2015-21 is obtained by regressing the residual AMOC component on the index of the residual AMOC component at 26.5°N. The units are $Sv$ ($10^6\ m^3 sec^{-1}$) per decade.

the AMOC during the historical period considered (Methods and Supplementary Tables 1–3). This and other inconsistencies between OMIP2 and the climate models used and the biases in those models make it challenging to further investigate physical processes involving the natural and externally forced interdecadal evolutions of the AMOC. Another important limitation of this study is that the historical AMOC field derived from OMIP2 is not the actual observation, but a proxy that cannot be easily validated since there is no direct continuous observation prior to April 2004. For instance, previous modeling studies do not agree on the interdecadal evolution of the AMOC in response to the excessive deep convection in the Labrador Sea in the 1990s[26,27,29]. It is also worthwhile to point out that the impact of anthropogenic aerosol on the AMOC is relatively unclear because of a large uncertainty in representing the indirect aerosol effect (i.e., aerosol-cloud radiative forcing) in climate models[17,76]. As such, a large uncertainty still remains in the time series of the AMOC derived from OMIP2 and the externally forced AMOC component derived from the climate

models used[17]. Due to the above-listed limitations, the conclusions presented in this study should be taken with caution.

There are some other limitations that can be addressed by additional modeling and analysis studies. In particular, it is implicitly assumed in this study that the externally forced and natural AMOC components are linearly separable and thus independent from each other. This is valid only because the externally forced AMOC component used in this study is the ensemble-mean AMOC field derived from many model realizations or multi-models, and thus is predetermined. However, in reality, the external forcing and natural atmosphere-ocean-ice variability interact with each other both linearly and non-linearly. It is also important to point out that the composite analysis derived from the CESM2 ensembles does not account for the actual amplitude of the residual AMOC component or the NAO-induced surface buoyancy flux; thus, not all aspects of the residual AMOC component and its decadal variation can be readily explained by the composite analysis. To address these limitations, we recommend three

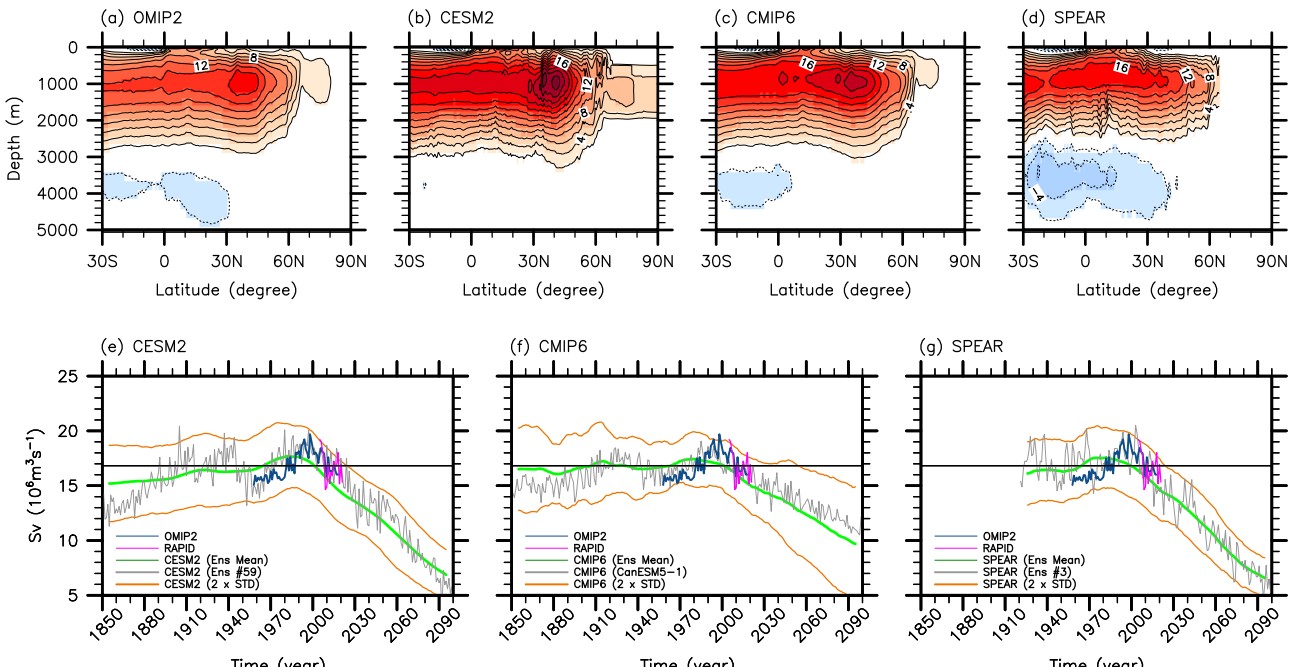

**Fig. 7 | Time-averaged AMOC and time series of the AMOC at 26.5°N. a–d** Time-averaged Atlantic Meridional Overturning Circulation (AMOC) transport stream-function for 1958 - 2018 derived from (**a**) the multi model-mean OMIP2, (**b**) the ensemble-mean CESM2, (**c**) the multi model-mean CMIP6, and (**d**) the ensemble-mean SPEAR. **e–g** The adjusted AMOC time series for OMIP2 (blue), RAPID (magenta), (**e**) the ensemble-mean CESM2 (green), (**f**) the multi model-mean CMIP6 (green), and (**g**) the ensemble-mean SPEAR (green). The gray lines in (**e**–**g**) indicate the adjusted AMOC at 26.5°N derived from (**e**) the CESM2 ensemble #59, (**f**) CanESM5-1, and (**g**) the SPEAR ensemble #3, examples of multiple realizations that contain both externally forced signal and natural variability (Methods). The orange lines in (**e**–**g**) indicate the 95% limits of (**e**) the adjusted CESM2 ensemble spread, (**f**) the adjusted CMIP6 model spread, and (**g**) the adjusted SPEAR ensemble spread (Methods), meaning that about 95% of the adjusted ensembles (or models) reside within the two orange lines. The units are Sv ($10^6\ m^3sec^{-1}$).

sets of specially designed ensemble ocean or fully coupled model experiments, with the first set forced with the observed NAO variability only, the second set with external forcing only, and the third set with both observed NAO variability and external forcing, similar to those described in ref. 40. Such experiments, carried out preferably using multiple high-resolution models, may allow us to address additional outstanding questions involving natural and externally forced ocean processes and the potential interactions between the two.

Lastly, it is worthwhile to briefly discuss a potential collapse of the AMOC suggested in several recent studies[77–79]. While such a cascade mechanism triggered by a substantial freshening in the subpolar North Atlantic is well founded by a sound theory[80], the current state-of-the-art climate models including those analyzed in this study do not show a complete shutdown of the AMOC during the 21st century[23]. However, it should be noted that future freshening of the North Atlantic due to the melting of the Greenland ice sheet and land-based glaciers and ice caps in the Arctic circle is not adequately represented in those models. Thus, it is important to continue improving climate models by incorporating realistic scenarios of the Greenland ice sheet melting to address this low likelihood-high impact possibility. Additionally, to further advance our capability to detect and attribute the historical AMOC and to predict the future AMOC, it is vital to continue monitoring the AMOC at multiple latitudes, advance our understanding of the relevant physical processes, and improve climate models and reconstructions of the historical AMOC.

## Methods

### RAPID & ERA5 datasets and OMIP2 models

The RAPID/Meridional Overturning Circulation and Heatflux Array/Western Boundary Time Series moored array (RAPID)[22] is used to obtain the observed time series of the AMOC at around 26.5°N from April 2004 to February 2022. To complement the RAPID time series and to extend our analysis to the period prior to April 2004, 10 surface-forced ocean and sea-ice models participating in the Ocean Model Intercomparison Project phase-2 (OMIP2[27], Supplementary Table 1) are also used to derive a time series of the AMOC profiles from 1958 to 2018, which is used here as a proxy for the observed AMOC of that period. Monthly anomalies of sea level pressure, derived from the European Center for Medium-Range Weather Forecasts (ECMWF) reanalysis-5 (ERA5)[56] are also used to explore the link between the AMOC and NAO.

### CESM2 & SPEAR large ensemble simulations and CMIP6 models

To explore the interdecadal evolution of the externally forced AMOC, we analyze two sets of large ensemble climate model simulations, based on the Community Earth System Model version 2 (CESM2)[36], and the Seamless System for Prediction and Earth System Research (SPEAR)[37], and 20 climate models participating in CMIP6 (Supplementary Tables 2 and 3). These models are used to derive the AMOC and monthly anomalies of sea level pressure for the CMIP6 historical period (1850 - 2014 for CESM2 and CMIP6, and 1921 - 2014 for SPEAR) as well as for the CMIP6 future period (2015 - 2100) under a medium-to-high end radiative forcing scenario of CMIP6 (i.e., Shared Socioeconomic Pathways 370; SSP-370) for CESM2 and CMIP6 and a high end radiative forcing scenario (SSP-585) for SPEAR. We use a total of 80 ensemble members of CESM2 initialized in 1850 with four different states of the AMOC (i.e., a strengthened, a decreasing, a weakened, and an increasing) from a preindustrial control simulation[36]. Similarly, we use a total of 30 ensemble members of SPEAR initialized in 1921[37].

### Adjusting OMIP2, CESM2, CMIP6 and SPEAR to observations

Figure 7a–d shows the time-averaged AMOC transport streamfunction during 1958–2018 derived from the multi-model mean OMIP2, the ensemble mean CESM2, the multi-model mean CMIP6, and the ensemble mean SPEAR. While the overall spatial structure of the AMOC is largely consistent between OMIP2 and CESM2, the maximum AMOC

transport at each latitude is distinctly larger in CESM2 compared to that in OMIP2. Specifically, the time-averaged AMOC transports at 26.5°N for OMIP2 and CESM2 are 15.8 Sv and 19.6 Sv, respectively (i.e., 3.8 Sv stronger in CESM2 than in OMIP2). For the period during which the direct observations (i.e., RAPID) and OMIP2 overlap (2005 - 2018), the time-averaged AMOC values are 16.9 Sv in RAPID and 15.9 Sv in OMIP2 (i.e., 1.0 Sv stronger in RAPID than in OMIP2).

To help compare the AMOC between OMIP2, CESM2, and RAPID, the AMOC time series of the multi-model mean OMIP2 is adjusted to match that of RAPID by adding the difference (i.e., 1.0 Sv). The AMOC time series of CESM2 at 26.5°N is also adjusted by subtracting 2.8 Sv (i.e., the difference between CESM2 and RAPID during 2005–2018). The CESM2 ensemble spread (i.e., standard deviation of 80 ensembles) of the annual mean AMOC during the observational period (2005 - 2021) is 0.7 Sv, whereas the standard deviation of the annual mean AMOC in RAPID is 1.4 Sv. Therefore, the CESM2 ensemble spread is multiplied by a factor of 2 to match the observed annual mean variability of the AMOC.

Figure 7e shows the adjusted annual mean AMOC time series at 26.5°N for OMIP2 (blue), CESM2 (green), and RAPID (magenta). Orange lines indicate the 95% limits of the adjusted CESM2 ensemble spread (i.e., 2 × standard deviation of ensembles × 2.0), meaning that about 95% of the adjusted 80 ensembles reside within the two orange lines. The gray line indicates the adjusted AMOC at 26.5°N derived from the CESM2 ensemble #59, an example of the 80 realizations that contain both the externally forced component and natural variability. The ensemble mean CESM2 shows a slow increase in the AMOC until around the mid-1980s, and then a persistent decline afterward, largely consistent with the multi-model mean of CMIP6 models[17,18]. On the other hand, the multi-model mean OMIP2 shows a relatively weakened state of the AMOC during 1958–1980, followed by a steep increase to the 1990s, and then a decline until 2018.

To allow a better visual comparison of the three interdecadal AMOC time series in Fig. 1a, the time-averaged RAPID AMOC transport value of 16.7 Sv for the observational period of 2005–2021 is removed from each time series in Fig. 1a. The observed AMOC value for the decade centered in 2020 (i.e., 2015–24) is approximated by averaging the AMOC values for only 7 years from 2015 to 21. Similarly, the OMIP2 AMOC value for the decade centered in 1960 (i.e., 1955 - 64) is approximated by averaging the AMOC values for only 7 years from 1958 to 1964. A similar adjustment procedure is applied to CMIP6 and SPEAR as shown in Fig. 7f and g, respectively. From 1958 to 2018, the time-averaged AMOC values in CMIP6 and SPEAR, prior to the adjustment, are 19.1 Sv and 16.7 Sv, respectively. Note that the multi-model ensemble spread of CMIP6 contains both natural variability and inter-model differences; thus, it is not adjusted to the observed annual mean variability of the AMOC.

### Computation of December-May NAO index
Sea level pressure anomalies from ERA5 are used to derive the December–May NAO index time series, which is measured as the difference in sea level pressure anomalies between the Azores high (36° – 40°N and 28°W – 20°W) and Icelandic low regions (63°–70°N and 25°–16°W). See Supplementary Note 3 and Supplementary Fig. 6 for more discussion on the sensitivity of the NAO index to the choice of datasets and computation methods. The same procedure is used to compute the December-May NAO index for each of the climate models used. For CESM2 and SPEAR, the corresponding ensemble-mean December-May NAO index is removed from the December-May NAO index for each ensemble member.

### CMIP6 models under the preindustrial scenario
In addition to the 20 CMIP6 models analyzed for the historical and future periods, we also analyze 22 CMIP6 models under the pre-industrial scenario with the fixed preindustrial $CO_2$ concentration of 280 ppm. Each of these model runs is a 500-year long, and used to carry out the lead-lag correlation analysis between the December-May NAO index and the residual AMOC component at 26.5°N, as well as the autocorrelation analysis of the residual AMOC component at 26.5°N. To derive monthly NAO time series, the difference in sea level pressure anomalies between the Azores high (36° – 40°N and 28 W° – 20°W) and Icelandic low regions (63° – 70°N and 25° – 16°W) is used. For each CMIP6 model, the long-term climatology is removed from the monthly NAO index. Supplementary Table 3 provides a summary of the 22 CMIP6 models under the preindustrial scenario.

## Data availability
The CESM2 data were downloaded from the NCAR Climate Data Gateway at https://www.earthsystemgrid.org. The CMIP6 data were downloaded from the Earth System Grid Federation portals at https://esgf-node.llnl.gov/projects/cmip6, https://esgf-node.ipsl.upmc.fr/projects/cmip6-ipsl, https://esgf-data.dkrz.de/projects/cmip6-dkrz, and https://esgf-index1.ceda.ac.uk/projects/cmip6-ceda. The SPEAR data were downloaded from NOAA's Geophysical Fluid Dynamics Laboratory at https://www.gfdl.noaa.gov/spear_large_ensembles. The RAPID data were downloaded from the RAPID AMOC monitoring page at https://rapid.ac.uk/rapidmoc/rapid_data/datadl.php. The ERA5 surface flux data were downloaded from ECMWF at https://www.ecmwf.int/en/forecasts/dataset/ecmwf-reanalysis-v5.

## Code availability
The analysis presented in this study is carried out using the NCAR Command Language (NCL). The NCL code is publicly available from the National Center for Atmospheric Research at https://www.ncl.ucar.edu/Download.

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

## Acknowledgements

The authors acknowledge Tom Delworth for helpful discussions and comments and for assisting the SPEAR data retrieval, and Lei Huang for carefully reviewing the manuscript. This work was supported by the base funding of NOAA's Atlantic Oceanographic and Meteorological Laboratory (AOML), by NOAA's Climate, Ecosystems, and Fisheries Initiative (CEFI), and by NOAA's Climate Program Office's Modeling, Analysis, Predictions, and Projections program. D.K. and D.L.V. were supported by NOAA's AOML in part under the auspices of the Cooperative Institute for Marine and Atmospheric Studies (CIMAS), a cooperative institute of the University of Miami and NOAA, cooperative agreement NA20OAR4320472. F.A.G. was supported by the Northern Gulf Institute under NOAA cooperative agreement NA21OAR4320190. D.L.V. was also supported by NOAA's Climate Variability and Predictability program grant NA20OAR4310407 and the Western Boundary Time Series program. S.Y. acknowledges support from the National Science Foundation (NSF) grant OCE-2406511. The National Center for Atmospheric Research (NCAR) is a major facility sponsored by NSF under Cooperative Agreement 1852977.

## Author contributions

S-K.L. conceived the study, performed the analysis and wrote the initial draft of the paper. D.K. assisted the analysis. All authors (S.-K.L., D.K., F.A.G., H.L., D.L.V., S.D., R.L., and S.Y.) significantly contributed to the discussion and interpretation of results, and reviewed and edited the paper.

## Competing interests

The authors declare no competing interests.
