## [Transparent Peer Review file · Nature Communications]

A pause in the weakening of the Atlantic meridional overturning circulation since the early 2010s

Corresponding Author: Dr Sang-Ki Lee

Version 0:

Reviewer comments:

Reviewer #1

(Remarks to the Author)

This study sheds light on the interdecadal variations of the Atlantic Meridional Overturning Circulation (AMOC) at 26.5°N. The study's utilization of a comprehensive dataset from CESM2, CMIP6, SPEAR, and ERA5 allows for robust analysis. This multi-model approach strengthens the credibility of the findings compared to relying on a single model. Separating the AMOC into externally forced and residual components is a significant contribution. It facilitates a clearer understanding of how external factors (like greenhouse gases) and internal variability (like the NAO) influence AMOC changes. I recommend the paper to be accepted after addressing the following comments.

1, A previous study (Sun et al. 2021) evaluated five representative indicators of the Atlantic Meridional Overturning Circulation (AMOC), including one atmospheric index based on accumulated atmospheric forcing and four oceanographic indices using surface and subsurface oceanographic variables in the North Atlantic Ocean. The AMOC_NAO index is an atmospheric index based on accumulated atmospheric forcing from the North Atlantic Oscillation (NAO). It serves as a proxy to represent the strength and variability of the AMOC, capturing the influence of atmospheric conditions on ocean circulation. The study mentions that from 2004 to 2010, the AMOC weakened, but from 2010 onwards, there has been a slight recovery. The AMOC_NAO index has been shown to be consistent with the low-frequency changes in the AMOC as measured by the RAPID monitoring array. The index accurately reflects the weakening trend observed from 2004 to 2010 and the subsequent slight recovery since 2010. This prior study is very relevant to the present study.

2, While the study explores the potential predictability of the residual AMOC state to predict its short-term evolution, it falls short of providing a definitive near-term outlook on the AMOC index. The authors could consider develop a more robust prediction method or model that incorporate the Residual AMOC, Lead-Lag Relationships and the external forced component. The residual component, potentially influenced by the NAO, should be explicitly integrated into the prediction. The lead-lag relationship between the residual AMOC and the NAO index should be factored into the prediction model. This could allow for predictions of future residual AMOC states based on current changes in the NAO. The influence of external forcing factors, such as greenhouse gas emissions, on the AMOC must also be considered in prediction models. The authors could consider different levels of anthropogenic forcing in the coming decades and their impact on the external forced component. By incorporating these elements, the study could provide a more robust prediction of the AMOC's near-term future.

3, The study does not explicitly account for the uncertainty associated with the data source and definition method of the NAO index. The authors should acknowledge the inherent uncertainty associated with the NAO index. This could involve quantifying the uncertainty arising from different data sources and definition methods. To assess how sensitive the study's findings are to the choice of NAO index, the authors could conduct a sensitivity analysis. This would involve re-running the analyses using different NAO index datasets. By comparing the results, the authors can evaluate the robustness of the conclusions and identify potential biases introduced by specific NAO index choices.

4, The residual component likely incorporates internal variability from sources other than the NAO. The study focuses on the NAO, but atmospheric circulation variability over the North Atlantic as a whole likely influences the AMOC through changes in wind patterns and surface heat fluxes. The authors could further investigate the contributions of these other atmospheric circulation modes to the residual AMOC component. Statistical techniques, such as multivariate analysis, can be employed to identify the relative contributions of different internal variability modes (including the NAO and others) to the residual

AMOC. By expanding the investigation beyond the NAO, future research can provide a more comprehensive picture of the internal factors that shape the residual component of AMOC's variations.

Reference:

Sun C, Zhang J, Li X, et al. Atlantic Meridional Overturning Circulation reconstructions and instrumentally observed multidecadal climate variability: A comparison of indicators[J]. International Journal of Climatology, 2021, 41(1): 763-778.

Reviewer #2

(Remarks to the Author)

Review of "A pause in the weakening of the Atlantic meridional overturning circulation since the early 2010s" by Lee et al.

This paper presents an analysis of the change in AMOC over the past decades, partitioning it into the externally forced (aerosols & ghgs) and residually forced (mainly from NAO). The paper uses observations, forced ocean models and climate models to perform the analysis and so is well supported; the methodology is sound and well-described. I thought the paper was very interesting and quantifies an oft mentioned but rarely quantified separation of variability in AMOC.

Major comments:

The discussion could address tipping points as the discussion is already about autocorrelation and this is the main indicator of the AMOC approaching tipping points. How much would this NAO-forced variability impact the assessment of increased autocorrelation.

The independence of NAO-driven variability from the 'external' forcings is an important point. Obviously there is a subtlety between natural and non-natural external forcing. Could these be interacting with one another? How would you tell if they were not?

Minor comments:

Sup fig 1c is brilliant. I think this is worth inclusion in the main text.

What is the ensemble selection included in the supp mat? This wasn't discussed in the text.

I25: be clear that this is a multi-model mean

I27: I think observational records show resilience over the past *four decades*.

I61: you should mention the specific program of AMOC observation as there are now a number of these.

I108: given the known dependence of AMOC decline on mean AMOC strength, it would be worth stating the mean values of each of these estimates here.

Fig. 3(top): Colorbar on NAO is not intuitive. Suggest sticking to one colour.

I203: the phrasing here is confusing. When you refer to "NAO in 2010", you mean a decadal average? Maybe this could be written as "NAO in the 2010s" or some other more accurate way. NAO in 2010 was famously negative and so this could cause confusion.

I251: a time integrated & delayed response makes sense. Does this mean that a correlation with the rate of change of NAO would not be lagged? Is there evidence of a quadrature lag between AMOC and NAO cycles?

Fig. 5 g-i is an interesting way to plot the autocorrelation. Can other estimators of autocorrelation also be included.

I324: probably no need for the 'obviously!' You are showing the value of the autocorrelation for understanding sources of predictability so maybe more complementary to decadal prediction systems than this sentence implies.

Supp fig 4. Typo in titles: Postive -> Positive

Reviewer #3

(Remarks to the Author)

General remarks:

This manuscript investigates the attribution of interdecadal AMOC change during the historical period to external forcing (mostly anthropogenic forcing) and nature variability by analyzing multiple sets of model simulations. The authors argue that the pause in the weakening of the AMOC since the early 2010s is a result of the competition between anthropogenic forcing-associated weakening and North Atlantic Oscillation-associated strengthening. Based on these results, they further predict a stronger AMOC (relative to 2005-2014) in the next several years. The manuscript is very well written and easy to understand and has appropriate caveats. While I find the manuscript is of high interest and holds promise to better our understanding of interdecadal AMOC variation and its predictability, I do have several concerns/comments, especially regarding the model uncertainty as listed below. I would like to suggest a major revision based on its current form.

Major concerns:

1. As to the historical AMOC variation, one outstanding issue is that the reversal from strengthening to weakening in historical AMOC happens in the 1980s in CMIP6 multi-model ensemble mean but 2000s in low-resolution OMIP2. On the other hand, the decreasing of anthropogenic aerosol loading starts from 1980s in the Northern Hemisphere (except Asia; Hoesly et al., 2018; Haustein et al., 2019). While nature variability can be a possibility for the lag of ~20 years in OMIP2, there are other possibilities. For example, the ECCOv4 ocean state estimate (Fig.1a in Zhu et al., 2023) and high-resolution OMIP2 (Fig.15a in Chassignet et al., 2020) produce an AMOC weakening in the early 1990s. Therefore, the attribution of AMOC variation is not exclusive and likely to be model dependent. Can the authors expand more on this?

References:

Hoesly, R. M. et al. Historical (1750–2014) anthropogenic emissions of reactive gases and aerosols from the Community Emissions Data System (CEDS). *Geosci. Model Dev.* 11, 369–408 (2018).

Haustein, K. et al. A Limited Role for Unforced Internal Variability in Twentieth-Century Warming. *J. Clim.* 32, 4893–4911(2019).

Zhu, C. et al. Likely accelerated weakening of Atlantic Overturning Circulation Emerges in Optimal Salinity Fingerprint. *Nature Communications*, 14, 1245 (2023).

Chassignet, E. P. et al. Impact of horizontal resolution on global ocean–sea ice model simulations based on the experimental protocols of the Ocean Model Intercomparison Project phase 2 (OMIP-2). *Geosci. Model Dev.*, 13, 4595–4637 (2020).

2. Low-resolution models usually produce lower AMOC intensity during the historical period considered in this study while high-resolution models simulate AMOC more like observations (Chassignet et al., 2020, Hirschi et al., 2020).

Considering the above two concerns, I would therefore like to encourage the authors to check the high-resolution CMIP6 and OMIP2 to see if the attribution differs from that in the low-resolution ones.

References:

Hirschi, J. J.-M., Barnier, B., Böning, C., Biastoch, A., Blaker, A. T., Coward, A., et al. The Atlantic meridional overturning circulation in high-resolution models. *Journal of Geophysical Research: Oceans*, 125, e2019JC015522 (2020).

3. The authors argue that subtropical AMOC is influenced by time-integrated NAO variability (in contrast to subpolar AMOC), which is physically possible. So why not show the correlation between AMOC and time-integrated NAO instead of NAO index?

Minor Comments:

4. Why the authors adjust annual variation of the AMOC while the focus of the study is interdecadal variation?

5. In several places of the text, the difference between total and externally force AMOC is referred to as residual AMOC. Residual AMOC is often used for describing total AMOC which is the sum of Eulerian AMOC and eddy-driven AMOC. I therefore suggest to rephrase it as residual component of the AMOC or the residual AMOC component.

6. Please provide the horizontal and vertical resolution for models in supplementary tables or Methods.

Version 1:

Reviewer comments:

Reviewer #1

(Remarks to the Author)

The authors have responded to my comments and made revisions accordingly. I recommend the publication of this paper.

Reviewer #3

(Remarks to the Author)

This is my re-review of Lee et al. I thank the authors for their efforts (especially the additional analysis of high-resolution models) in addressing my concerns. I think the manuscript has been much improved by considering the three reviewers' comments. Therefore, I would like to recommend it for acceptance.

Reviewer #1 (Remarks to the Author):

This study sheds light on the interdecadal variations of the Atlantic Meridional Overturning Circulation (AMOC) at 26.5°N. The study's utilization of a comprehensive dataset from CESM2, CMIP6, SPEAR, and ERA5 allows for robust analysis. This multi-model approach strengthens the credibility of the findings compared to relying on a single model. Separating the AMOC into externally forced and residual components is a significant contribution. It facilitates a clearer understanding of how external factors (like greenhouse gases) and internal variability (like the NAO) influence AMOC changes. I recommend the paper to be accepted after addressing the following comments.

>

We greatly appreciate helpful and insightful comments. We have revised the manuscript by addressing the reviewer's suggestions and comments. We believe that the manuscript is much improved after addressing the points raised by the reviewer. Our replies (in black colored fonts) are shown below for each of the major and minor comments.

1, A previous study (Sun et al. 2021) evaluated five representative indicators of the Atlantic Meridional Overturning Circulation (AMOC), including one atmospheric index based on accumulated atmospheric forcing and four oceanographic indices using surface and subsurface oceanographic variables in the North Atlantic Ocean. The AMOC_NAO index is an atmospheric index based on accumulated atmospheric forcing from the North Atlantic Oscillation (NAO). It serves as a proxy to represent the strength and variability of the AMOC, capturing the influence of atmospheric conditions on ocean circulation. The study mentions that from 2004 to 2010, the AMOC weakened, but from 2010 onwards, there has been a slight recovery. The AMOC_NAO index has been shown to be consistent with the low-frequency changes in the AMOC as measured by the RAPID monitoring array. The index accurately reflects the weakening trend observed from 2004 to 2010 and the subsequent slight recovery since 2010. This prior study is very relevant to the present study.

>

Thank you very much for bringing our attention to Sun et al. (2021). In the revised manuscript, this paper along with Mecking et al. (2014), McCarthy et al. (2015) and Smeed et al. (2014) is now cited. We have clarified that the link between the time integral of NAO and AMOC was proposed by Sun et al. (2021) and others.

References

Mecking, J.V., Keenlyside, N.S. & Greatbatch, R.J. Stochastically-forced multidecadal variability in the North Atlantic: a model study. *Clim Dyn* 43, 271–288 (2014).
<https://doi.org/10.1007/s00382-013-1930-6>

Smeed, D. A. et al. Observed decline of the Atlantic meridional overturning circulation 2004–2012. *Ocean Sci.*, 10, 29–38 (2014).

McCarthy, G., Haigh, I., Hirschi, JM. et al. Ocean impact on decadal Atlantic climate variability revealed by sea-level observations. *Nature* 521, 508–510 (2015).
<https://doi.org/10.1038/nature14491>

Sun C, Zhang J, Li X, et al. Atlantic Meridional Overturning Circulation reconstructions and instrumentally observed multidecadal climate variability: A comparison of indicators. *Int J Climatol*. 2021; 41: 763–778. <https://doi.org/10.1002/joc.6695>

2, While the study explores the potential predictability of the residual AMOC state to predict its short-term evolution, it falls short of providing a definitive near-term outlook on the AMOC index. The authors could consider develop a more robust prediction method or model that incorporate the Residual AMOC, Lead-Lag Relationships and the external forced component. The residual component, potentially influenced by the NAO, should be explicitly integrated into the prediction. The lead-lag relationship between the residual AMOC and the NAO index should be factored into the prediction model. This could allow for predictions of future residual AMOC states based on current changes in the NAO. The influence of external forcing factors, such as greenhouse gas emissions, on the AMOC must also be considered in prediction models. The authors could consider different levels of anthropogenic forcing in the coming decades and their impact on the external forced component. By incorporating these elements, the study could provide a more robust prediction of the AMOC's near-term future.

>

We appreciate this suggestion. Based on a simple auto-regression model of order 1 (AR1) applied to CESM2, we derived a range of the residual AMOC at 26.5N expected during the next 1-5 years. Using the residual AMOC value of +2.2 Sv for 2015-21, the expected value of the residual AMOC during 2022-26 is about $+1.7 \pm 1.1$ Sv, suggesting that the residual AMOC may wane down by -0.5 ± 1.1 Sv. Combining this with the externally-forced weakening of -0.6 Sv from 2015-21 to 2022-26, the total AMOC is expected to weaken by -1.1 ± 1.1 Sv in 2022-26 compared to that in 2015-21. Thus, using the observed AMOC value of 16.6 Sv in 2015-21, the expected total AMOC in 2022-26 is 15.5 ± 1.1 Sv. We now include Supplementary Table 4 that summarizes our prediction of the AMOC in 2022-26 based on the simple auto-regression analysis of CESM2, CMIP6 and SPEAR. The manuscript is also updated in section “Potential decadal predictability of the residual AMOC component”.

>

Improving the multi-year prediction using other potential predictors such as the NAO could be beneficial. It can be achieved by using a higher order AR model and a weighted time-integral of the NAO index as proposed in Mecking et al. (2014). However, we feel that developing such a more complex statistical prediction model is beyond the scope of this study. We will definitely keep it (i.e., developing a statistical AMOC prediction model) in mind as a potential future work. Additionally, we think that decadal dynamic forecast systems with realistic initializations of the ocean are better suited for the actual decadal prediction of the AMOC. It is also worthwhile to point out that even those decadal dynamic forecast models display large differences between them. Thus, the World Meteorological Organization predicted “the total AMOC around

30°N is likely to be near or slightly below the recently observed values” without providing the exact value of the predicted AMOC.

Supplementary Table 4. Expected changes in the AMOC components during 2022-26.

Expected changes in the AMOC components (i.e., residual, externally forced, and total) from 2015-21 to 2022-26 derived from CESM2, CMIP6 and SPEAR. The residual AMOC component values for 2022-26 (i.e., 1.7, 1.0, and 1.0 Sv for CESM2, CMIP6 and SPEAR, respectively) are derived based on a simple autoregression tool (i.e., Figures 5g-i) applied to the residual AMOC component values for 2015-21 (i.e., 2.2, 1.3, and 1.9 Sv for CESM2, CMIP6 and SPEAR, respectively). Error estimates are based on the 95% limits of the regressed residual AMOC component values. The expected total AMOC values for 2022-26 based on the observed value of the AMOC during 2015-21 (i.e., 16.6 Sv) are also indicated (values inside parentheses). The units are Sv ($10^6 m^3 sec^{-1}$).

Model \ AMOC	Estimated Changes in Residual AMOC Component at 26.5°N	Estimated Changes in Externally forced AMOC at 26.5°N	Estimated Changes in Total AMOC at 26.5°N
CESM2	-0.5 ± 1.1	-0.6	-1.1 ± 1.1 (15.5 ± 1.1)
CMIP6	-0.3 ± 1.1	-0.4	-0.7 ± 1.1 (15.9 ± 1.1)
SPEAR	-0.9 ± 1.3	-0.3	-1.2 ± 1.3 (15.4 ± 1.3)

3, The study does not explicitly account for the uncertainty associated with the data source and definition method of the NAO index. The authors should acknowledge the inherent uncertainty associated with the NAO index. This could involve quantifying the uncertainty arising from different data sources and definition methods. To assess how sensitive the study's findings are to the choice of NAO index, the authors could conduct a sensitivity analysis. This would involve re-running the analyses using different NAO index datasets. By comparing the results, the authors can evaluate the robustness of the conclusions and identify potential biases introduced by specific NAO index choices.

> In this study, the difference in sea level pressure anomalies between the Azores high (36° – 40°N and 28W°28°W – 20°W) and Icelandic low regions (63° – 70°N and 25° – 16°W) derived from ERA5 is used to compute December-May NAO time series. To examine the sensitivity of the NAO index to the choice of datasets and computation methods, we derive several alternative sets of December-May NAO time series using sea level pressure anomalies from the National Center for Environmental Prediction - National Center for Atmospheric Research Reanalysis -1 (NCEP1), and the Japanese 55-year Reanalysis (JRA55), and applying an Empirical Orthogonal Function (EOF) analysis (e.g., Hurrell and Deser, 2009) to sea level pressure anomalies in the North Atlantic (0° - 80°N and 90°W - 0°).

> As summarized in Supplementary Figure 6, the spatiotemporal NAO patterns derived from NCEP1 and JRA55 and those computed using the EOF-based definition the NAO are largely consistent with those derived from ERA5 and the area-averaged definition of the NAO. Some noticeable differences are seen only in the EOF-based ERA5 NAO time series prior to the late 1960s and also around the mid-to-late 1990s to a lesser degree.

The supporting figure and related discussion are now added in the supplementary information (Supplementary Note 3).

>

Hurrell, J. W. & Deser, C. North Atlantic climate variability: The role of the North Atlantic Oscillation. *J. Mar. Syst.* 78, 28-41 (2009).

Supplementary Figure 6. Spatiotemporal patterns of the area-averaged and EOF-based December-May NAO. (top panels) Spatial patterns of the area-averaged December-May NAO derived from (left) ERA5, (middle) NCEP1 and (right) JRA55. (middle panels) Spatial patterns of the EOF-based December-May NAO derived from (left) ERA5, (middle) NCEP1 and (right) JRA55. (bottom panel) 5-year running-mean time series of the area-averaged (solid lines) and EOF-based (dashed lines) December-May NAO derived from ERA5 (blue), NCEP1 (red), and

JRA55 (green). The units for sea level pressure anomalies in the upper and middle panels are *hPa* per one standard deviation of the NAO.

4, The residual component likely incorporates internal variability from sources other than the NAO. The study focuses on the NAO, but atmospheric circulation variability over the North Atlantic as a whole likely influences the AMOC through changes in wind patterns and surface heat fluxes. The authors could further investigate the contributions of these other atmospheric circulation modes to the residual AMOC component. Statistical techniques, such as multivariate analysis, can be employed to identify the relative contributions of different internal variability modes (including the NAO and others) to the residual AMOC. By expanding the investigation beyond the NAO, future research can provide a more comprehensive picture of the internal factors that shape the residual component of AMOC's variations.

>

We appreciate this suggestion. In addition to the NAO, there are other coherent atmospheric patterns affecting the North Atlantic Ocean such as Eastern Atlantic Pattern (EAP), Scandinavia Pattern, and Pacific-North American Pattern. Msadek & Frankignoul (2009) showed that EAP is significantly correlated with the AMOC when the EAP leads by 10~20 years while the NAO is not correlated with the AMOC in a long-term climate simulation based on the IPSL-CM4 climate model. Ruprich-Robert & Cassou (2015), on the other hand, showed that both the NAO and EAP may lead the AMOC by 30~40 years in a long-term climate simulation based on the CNRM-CM5 model. Hence, the relationship between the EAP and AMOC appears to be model-dependent. More importantly, the mechanical link between the EAP and AMOC has not been demonstrated using observations. Thus, we think that additional study is needed to clarify the physical link between the EAP and AMOC and to demonstrate if such link actually exists by using observational data. On the other hand, NAO-like SLP anomalies largely dominate the spatial patterns of the decadal sea level pressure anomalies as shown in Figure 4. Combining this with the weak evidence of the EAP-AMOC relationship, we feel that addressing the potential role of EAP and other atmospheric patterns on the interdecadal AMOC variability is beyond the scope of this study.

>

Nevertheless, we agree that the NAO is not the only factor that influences the residual AMOC. So, in the revised discussion, we added a brief discussion on this topic, and stated that further studies are needed to identify the role of EAP and other atmospheric variability on the residual AMOC.

Msadek, R., & Frankignoul, C. (2009). Atlantic multidecadal oceanic variability and its influence on the atmosphere in a climate model. *Climate Dynamics*, 33(1), 45–62. <https://doi.org/10.1007/s00382-008-0452-0>

Ruprich-Robert, Y., & Cassou, C. (2015). Combined influences of seasonal East Atlantic Pattern and North Atlantic Oscillation to excite Atlantic multidecadal variability in a climate model. *Climate Dynamics*, 44(1–2), 229–253. <https://doi.org/10.1007/s00382-014-2176-7>

Reference:

Sun C, Zhang J, Li X, et al. Atlantic Meridional Overturning Circulation reconstructions and instrumentally observed multidecadal climate variability: A comparison of indicators[J]. *International Journal of Climatology*, 2021, 41(1): 763-778.

Reviewer #2 (Remarks to the Author):

Review of “A pause in the weakening of the Atlantic meridional overturning circulation since the early 2010s” by Lee et al.

This paper presents an analysis of the change in AMOC over the past decades, partitioning it into the externally forced (aerosols & ghgs) and residually forced (mainly from NAO). The paper uses observations, forced ocean models and climate models to perform the analysis and so is well supported; the methodology is sound and well-described. I thought the paper was very interesting and quantifies an oft mentioned but rarely quantified separation of variability in AMOC.

>

We greatly appreciate insightful comments. We have revised the manuscript by addressing the reviewer’s suggestions and comments. In particular, we added a discussion on the potential collapse of the AMOC and the future research needed to address this important topic. By addressing key points raised by the reviewer, we believe that the manuscript is much improved. Our replies (in black colored fonts) are shown below for each of the major and minor comments.

Major comments:

The discussion could address tipping points as the discussion is already about autocorrelation and this is the main indicator of the AMOC approaching tipping points. How much would this NAO-forced variability impact the assessment of increased autocorrelation.

>

Thank you for this suggestion. First, here is a brief summary of our understanding on this topic. A recent study suggested a potential collapse of the AMOC occurring around the mid-21st century under the current level of greenhouse gas emissions (Ditlevsen & Ditlevsen, 2023). The physical basis for a potential AMOC collapse was first brought up by Henry Stommel (Stommel, 1961). Specifically, sufficiently strong freshwater forcing (e.g., ice sheet melting from Greenland’s glaciers due to increasing anthropogenic carbon) in the subpolar North Atlantic may decrease the AMOC, which may in turn limit the salt transport from the subtropical North Atlantic to the sinking site (e.g., Labrador and Irminger Seas), further reducing the salinity in the subpolar North Atlantic. This positive feedback may lead to a tipping point at which the AMOC may shut down completely. Stommel (1961) demonstrated this cascade mechanism using simple temperature and salinity equations and the equation of state. While Stommel’s study supports the possibility of a complete shutdown of the AMOC in the future, the current state-of-the-art climate models do not agree with Ditlevsen & Ditlevsen (2023). The majority of CMIP6 models point to a weakening of the AMOC by 34 ~ 45% of its present-day strength by 2100, while none of those models show a complete shutdown of the AMOC (Weijer et al., 2020).

>

In Ditlevsen & Ditlevsen (2023) and other studies (e.g., Boers, 2021), increasing AMOC variance and lag-1 autocorrelation were used as warning signals for a collapse of the

AMOC. However, we feel that it is beyond the scope of this study to go further onto discussing these warning signals in this manuscript. Instead, we added a brief discussion on the potential collapse of the AMOC in the revised discussion, citing Ditlevsen & Ditlevsen (2023) and others as described below

>

It is worthwhile to briefly discuss a potential collapse of the AMOC suggested in several recent studies (i.e., Boers, 2021; Ditlevsen & Ditlevsen, 2023; van Westen et al., 2024). While such a cascade mechanism triggered by a substantial freshening in the subpolar North Atlantic is well founded by a sound theory (Stommel, 1961), the current state-of-the-art climate models including those analyzed in this study do not show a complete shutdown of the AMOC during the 21st century (Weijer et al., 2020). However, it should be noted that future freshening of the North Atlantic due to the melting of the Greenland ice sheet and land-based glaciers and ice caps in the Arctic circle is not adequately represented in those models. Thus, it is important to continue improving models by incorporating realistic scenarios of the Greenland ice sheet melting to address this low likelihood-high impact possibility.

>

On the issue of the NAO and AMOC autocorrelation, it is unclear to what extent the NAO contributes to the autocorrelation of the AMOC (at 10-year lag for instance). We can speculate that the NAO likely reduces the autocorrelation of the AMOC (at 10-year lag for instance) since the NAO is a prime source of high-frequency variability (i.e., noise) of the AMOC. However, to fully explore that, we may need to carry out a series of model experiments with the NAO suppressed. We feel that such additional analysis is beyond the main scope of this study.

References

Boers, N. Observation-based early-warning signals for a collapse of the Atlantic Meridional Overturning Circulation. *Nat. Clim. Chang.* 11, 680–688 (2021).

<https://doi.org/10.1038/s41558-021-01097-4>

Ditlevsen, P., Ditlevsen, S. Warning of a forthcoming collapse of the Atlantic meridional overturning circulation. *Nat Commun* 14, 4254 (2023). <https://doi.org/10.1038/s41467-023-39810-w>

van Westen, R. M., Kliphuis, M. & Dijkstra, H. A. Physics-based early warning signal shows that AMOC is on tipping course. *Sci. Adv.* 10, eadk1189.

<https://doi.org/10.1126/sciadv.adk1189> (2024).

Stommel, H. Thermohaline Convection with Two Stable Regimes of Flow. *Tellus*, 13: 224-230. <https://doi.org/10.1111/j.2153-3490.1961.tb00079.x> (1961)

The independence of NAO-driven variability from the 'external' forcings is an important point. Obviously there is a subtlety between natural and non-natural external forcing. Could these be interacting with one another? How would you tell if they were not?

>

This is one of the major limitations in our model-based analysis study. We agree with the reviewer that we cannot separate them cleanly because they may interact with one another. This is why we propose (in the discussion section) a follow-up study to carry out three sets of specially designed ensemble ocean or fully coupled model experiments, with the first set forced with the observed NAO variability only, the second set with external forcing only, and the third set with both observed NAO variability and external forcing, similar to those described in Delworth et al. (2016). Such experiments, carried out using multiple high-resolution models, may allow us to address additional outstanding questions involving natural and externally forced ocean processes and the potential interactions between the two.

>

Delworth, T. et al. The North Atlantic Oscillation as a driver of rapid climate change in the Northern Hemisphere. *Nature Geosci.* **9**, 509–512 (2016).

Minor comments:

Sup fig 1c is brilliant. I think this is worth inclusion in the main text.

>

Thank you for this suggestion. Supplementary Figure 1 along with Supplementary Figures 2 and 3 are now moved to the main text as Figure 7.

What is the ensemble selection included in the supp mat? This wasn't discussed in the text.

>

The gray line in Supplementary Figures 1c (now Figure 7d) shows the adjusted AMOC from CESM2 ensemble #59. The gray line in Supplementary Figure 2c (now Figure 7e) shows the adjusted AMOC from CanESM5-1, which is one of the 20 CMIP6 model simulations used in this study. The gray line in Supplementary Figure 3c (now Figure 7f) shows the adjusted AMOC from SPEAR ensemble #3. They are shown as examples of individual members that contain both externally forced signal and natural variability as briefly mentioned in the figure captions and also in the method section.

I25: be clear that this is a multi-model mean

>

This sentence is revised to *The current state-of-the-art climate models when combined together suggest that the anthropogenic weakening of the Atlantic Meridional Overturning Circulation (AMOC) has already begun since the mid-1980s.*

I27: I think observational records show resilience over the past *four decades*.

>

We are not aware of any direct continuous AMOC observation record before 2004. Proxy records based on SST or other variables are not counted in this study. To be more precise, we changed “*direct observational records during the past two decades*” to “*direct continuous observational records during the past two decades*”.

I61: you should mention the specific program of AMOC observation as there are now a number of these.

>

This is where the key motivation for this study is building up. So, we prefer not to introduce the observational program in this sentence because the main message in this sentence could be sidetracked by spelling out the entire program name “the RAPID/Meridional Overturning Circulation and Heatflux Array/Western Boundary Time Series”. Additionally, RAPID is the only AMOC observational program at 26.5N and two references are also provided in this sentence for clarification.

I108: given the known dependence of AMOC decline on mean AMOC strength, it would be worth stating the mean values of each of these estimates here.

>

The mean AMOC values in OMIP2 and CESM2 are discussed in the method section. These values are adjusted as described there. Specifically, for the period during which OMIP2 and CESM2 overlap (2005 - 2018), the time-averaged AMOC transports at 26.5°N for OMIP2 and CESM2 are 15.8 Sv and 19.6 Sv, respectively. For the period during which the direct observations (i.e., RAPID) and OMIP2 overlap (2005 - 2018), the time averaged AMOC values are 16.9 Sv in RAPID and 15.9 Sv in OMIP2. Note that the time mean AMOC values of OMIP2 and CESM2 are adjusted.

>

In the revised method section, the time mean AMOC values of CMIP6 and SPEAR, prior to the adjustment, are indicated. Supplementary Tables 1-3 also provide the time mean AMOC value for CESM2, SPEAR and each of the OMIP2 and CMIP6 models used.

Fig. 3(top): Colorbar on NAO is not intuitive. Suggest sticking to one colour.

>

Done.

I203: the phrasing here is confusing. When you refer to "NAO in 2010", you mean a decadal average? Maybe this could be written as "NAO in the 2010s" or some other more accurate way. NAO in 2010 was famously negative and so this could cause confusion.

>

This is now changed to “NAO in 2010 (i.e., 2005-14)”.

I251: a time integrated & delayed response makes sense. Does this mean that a correlation with the rate of change of NAO would not be lagged? Is there evidence of a quadrature lag between AMOC and NAO cycles?

>

The idea of estimating the AMOC based on time-integrated NAO was discussed in several papers. For instance, Smeed et al. (2014) compared the time-integrated NAO with the AMOC to find a reasonable agreement between them. Smeed et al. (2012) and other three related papers (Mecking et al, 2014; McCarthy et al. 2015; Sun et al., 2020) are referenced in the revised manuscript.

Fig. 7 from Smeed et al. (2014). Annual average estimates of the AMOC from the 26N array (red, Sv right axis, error bar = 1.5 Sv), estimates of the AMOC from 6 hydrographic sections (black, Sv right axis, error bar = 5 Sv), the time series of annual average values of the AMO (blue, °C left axis) and accumulated NAO index (green, arbitrary units).

References

Mecking, J.V., Keenlyside, N.S. & Greatbatch, R.J. Stochastically-forced multidecadal variability in the North Atlantic: a model study. *Clim Dyn* 43, 271–288 (2014).

<https://doi.org/10.1007/s00382-013-1930-6>.

Smeed, D. A. et al. Observed decline of the Atlantic meridional overturning circulation 2004–2012. *Ocean Sci.*, 10, 29–38 (2014).

McCarthy, G., Haigh, I., Hirschi, JM. et al. Ocean impact on decadal Atlantic climate variability revealed by sea-level observations. *Nature* 521, 508–510 (2015).

<https://doi.org/10.1038/nature14491>

Sun C, Zhang J, Li X, et al. Atlantic Meridional Overturning Circulation reconstructions and instrumentally observed multidecadal climate variability: A comparison of indicators. *Int J Climatol.* 2021; 41: 763–778. <https://doi.org/10.1002/joc.6695>

Fig. 5 g–i is an interesting way to plot the autocorrelation. Can other estimators of autocorrelation also be included.

>

In the revised manuscript, we added the zero-crossing of the autocorrelation.

I324: probably no need for the 'obviously!' You are showing the value of the autocorrelation for understanding sources of predictability so maybe more complementary to decadal prediction systems than this sentence implies.

>

Done.

Supp fig 4. Typo in titles: Postive -> Positive

>

Fixed.

Reviewer #3 (Remarks to the Author):

General remarks:

This manuscript investigates the attribution of interdecadal AMOC change during the historical period to external forcing (mostly anthropogenic forcing) and nature variability by analyzing multiple sets of model simulations. The authors argue that the pause in the weakening of the AMOC since the early 2010s is a result of the competition between anthropogenic forcing-associated weakening and North Atlantic Oscillation-associated strengthening. Based on these results, they further predict a stronger AMOC (relative to 2005-2014) in the next several years. The manuscript is very well written and easy to understand and has appropriate caveats. While I find the manuscript is of high interest and holds promise to better our understanding of interdecadal AMOC variation and its predictability, I do have several concerns/comments, especially regarding the model uncertainty as listed below. I would like to suggest a major revision based on its current form.

>

We greatly appreciate the insightful and detailed comments. We have revised the manuscript by carefully addressing the reviewer's suggestions and comments. In particular, we carried out additional analysis using high resolution (~10 km) OMIP2 and medium resolution (~25km) CMIP6 models to address the model dependency issue. By addressing this and other key points raised by the reviewer, we believe that the manuscript is much improved. Our replies (in black colored fonts) are shown below for each of the major and minor comments.

Major concerns:

1. As to the historical AMOC variation, one outstanding issue is that the reversal from strengthening to weakening in historical AMOC happens in the 1980s in CMIP6 multi-model ensemble mean but 2000s in low-resolution OMIP2. On the other hand, the decreasing of anthropogenic aerosol loading starts from 1980s in the Northern Hemisphere (except Asia; Hoesly et al., 2018; Haustein et al., 2019). While nature variability can be a possibility for the lag of ~20 years in OMIP2, there are other possibilities. For example, the ECCOV4 ocean state estimate (Fig.1a in Zhu et al., 2023) and high-resolution OMIP2 (Fig.15a in Chassignet et al., 2020) produce an AMOC weakening in the early 1990s. Therefore, the attribution of AMOC variation is not exclusive and likely to be model dependent. Can the authors expand more on this?

References:

Hoesly, R. M. et al. Historical (1750–2014) anthropogenic emissions of reactive gases and aerosols from the Community Emissions Data System (CEDS). *Geosci. Model Dev.* 11, 369–408 (2018).

Haustein, K. et al. A Limited Role for Unforced Internal Variability in Twentieth-Century Warming. *J. Clim.* 32, 4893–4911(2019).

Zhu, C. et al. Likely accelerated weakening of Atlantic Overturning Circulation Emerges in Optimal Salinity Fingerprint. *Nature Communications*, 14, 1245 (2023).

Chassignet, E. P. et al. Impact of horizontal resolution on global ocean–sea ice model simulations based on the experimental protocols of the Ocean Model Intercomparison Project phase 2 (OMIP-2). *Geosci. Model Dev.*, 13, 4595–4637 (2020).

>

We agree that the attribution of the interdecadal AMOC variation presented in this study is not exclusive and to some extent model dependent. This issue of model dependency is addressed to some degree in this study by analyzing multiple OMIP2 and CMIP6 models along with two large ensemble simulations (CESM2 and SPEAR). To further explore this point, we carried out further analysis using additional OMIP2 and CMIP6 models

>

The multi model-mean OMIP2 indicates an increase in the total AMOC from 1958-64 to 1995-2004 followed by a decrease after the 1995-2004 peak (Figure 1). However, as shown in Supplementary Figure 1, large inter-model spread exists in the OMIP2 models. In particular, out of the total 10 OMIP2 models, six have the AMOC peak in 1985-94, three have the peak in 1995-2004, and one has the peak in 2005-14.

>

Previous studies reported that low resolution OMIP2 models tend to produce weaker than observed AMOC intensity while the AMOC intensity in high resolution OMIP2 models agree better with observations (e.g., Chassignet et al., 2020; Hirschi et al., 2020). Thus, we next explore if there are any systematic differences in the interdecadal AMOC swing between high and low resolution OMIP2 models. Specifically, we compare four sets of high (~10 km) and low (~100 km) resolution OMIP2 models discussed in Chassignet et al. (2020). As shown in Supplementary Figure 2, the high resolution version of the OMIP2 models tend to show higher amplitude interdecadal swing compared to the low resolution version models. However, both the high and low resolution versions of the four OMIP2 models display the AMOC peak in 1985-94 rather than 1995-2004, although inter-model spread is quite large (Supplementary Figure 3).

>

In summary, based on the results shown in Supplementary Figures 1-3, it appears that high-resolution OMIP2 models tend to show higher amplitude interdecadal AMOC swing. However, the timing of the AMOC peak (1985-94 or 1995-2004) in OMIP2 models is largely model dependent, and its dependency to model resolution is unclear. The supporting figures and related discussion are now added in the supplementary information (Supplementary Note 1). All suggested references are also cited.

>

ECCOv4 covers only the period of 1992–2017. So, we could not explicitly compare the ECCOv4 AMOC between 1985-94 and 1995-2004 periods.

Supplementary Figure 1. Decade-averaged time series of the AMOC anomalies at 26.5°N from 10 OMIP2 models. Decade-averaged time series of the AMOC anomalies at 26.5°N from the decade centered in 1960 (i.e., 1958-64) to the decade centered in 2010 (i.e., 2005-14), derived from ten OMIP2 models used in this study.

Supplementary Figure 2. Decade-averaged time series of the AMOC anomalies at 26.5°N in high and low resolution OMIP2 models. (a) Decade-averaged time series of the AMOC anomalies at 26.5°N from the decade centered in 1960 (i.e., 1958-64) to the decade centered in 2010 (2005-14), derived from four sets of high and low resolution OMIP2 models discussed in Chassignet et al. (2020). (b) Same as (a) except that the rate of interdecadal AMOC change is shown. The error bars in (a) indicate standard deviation from the ensemble-mean. Note that OMIP2 model runs are typically carried out for 366 years by repeating six cycles of the 61-year (1958-2018). However, for the high and low resolution OMIP2 simulations used in ref 1, no spin-up run was carried out. Hence, the AMOC time series during the first 17 years (i.e., 1958-1974) are stippled because they are contaminated by spin-up issues, and thus should be disregarded.

Supplementary Figure 3. Decade-averaged time series of the AMOC anomalies at 26.5°N in four sets of high and low resolution OMIP2 models. Decade-averaged time series of the AMOC anomalies at 26.5°N from the decade centered in 1960 (i.e., 1958-64) to the decade centered in 2010 (i.e., 2005-14), derived from four sets of high and low resolution OMIP2 models: (a) AWI-FESOM, (b) FSU-HYCOM, (c) IAP-LICOM, and (d) NCAR-POP. The rate of interdecadal AMOC change for each OMIP2 model is also shown in the lower panel. The AMOC time series during the first 17 years (i.e., 1958-1974) are stippled because they are contaminated by spin-up issues (Chassignet et al., 2020), and thus should be disregarded.

2. Low-resolution models usually produce lower AMOC intensity during the historical period considered in this study while high-resolution models simulate AMOC more like observations (Chassignet et al., 2020, Hirschi et al., 2020).

Considering the above two concerns, I would therefore like to encourage the authors to check the high-resolution CMIP6 and OMIP2 to see if the attribution differs from that in the low-resolution ones.

References:

Hirschi, J. J.-M., Barnier, B., Böning, C., Biastoch, A., Blaker, A. T., Coward, A., et al. The Atlantic meridional overturning circulation in high-resolution models. *Journal of Geophysical Research: Oceans*, 125, e2019JC015522 (2020).

>

It is now stated in the Supplementary Note 1 that low resolution models tend to produce weaker than observed AMOC intensity while the AMOC intensity in high-resolution models agree better with observations, referencing Chassignet et al. (2020) and Hirschi et al. (2020).

>

As discussed in our reply to reviewer comment #1, the high resolution version of some OMIP2 models tend to show higher amplitude interdecadal swing compared to the low resolution version (Supplementary Figures 2 & 3). However, the timing of the AMOC peak (1985-94 or 1995-2004) in those OMIP2 models is largely model dependent, and its dependency to model resolution is unclear. These additional conclusion points with respect to OMIP2 models are now discussed in Supplementary Note 1.

>

We next explore the potential effects of horizontal resolution on the externally forced interdecadal AMOC signal in CMIP6 models. To do so, it is important to use CMIP6 models with multiple ensemble members available in both high (or medium) and low resolutions. Thus, here we use the HadGEM3-GC31 simulations under the historical and SSP-585 scenarios (Jones et al. 2024), available in both low (~100 km) and medium (~25 km) resolutions with 4 ensemble members for both resolutions.

>

As shown in Supplementary Figure 4, the externally forced interdecadal AMOC signal in the low resolution runs (HadGEM3-GC31-LL) is much weaker than that in the medium resolution runs (HadGEM3-GC31-MM). The weaker externally forced AMOC signal in HadGEM3-GC31-LL appears to be more consistent with that of the multi model-mean CMIP6 (Figures 2a & 2b), whereas the stronger externally forced AMOC signal in HadGEM3-GC31-MM appears to be more consistent with that of the ensemble-mean CESM2 (Figures 1a & 1b) and the ensemble-mean SPEAR (Figures 2c & 2d). The externally forced AMOC in HadGEM3-GC31-LL displays its maximum in 1985-94 and a slightly weaker amplitude in 1975-84. In HadGEM3-GC31-MM, the externally forced AMOC displays its maximum in 1975-84 and a slightly weaker amplitude in 1985-94. This result is overall consistent with the externally forced AMOC signals in CESM2 (Figure 1a; the maximum in 1975-84 and 1985-94), CMIP6 (Figure 2a; the maximum in 1975-84) and SPEAR (Figure 2c; the maximum in 1975-84).

>

In summary, the additional analysis with the low and medium resolution versions of HadGEM3-GC31 indicates that the overall interdecadal evolutions of the externally forced and residual AMOC components in the low and medium resolution versions of HadGEM3-GC31 (Supplementary Figure 4) are largely consistent with those in CESM2, CMIP6 and SPEAR (Figures 1 and 2). However, the externally forced interdecadal AMOC signal is much weaker in the low resolution runs than that in the medium resolution runs. As such, the interdecadal swing of the residual AMOC component is also weaker in the low resolution runs than that in the medium resolution runs

(Supplementary Figure 5). Therefore, our analysis with additional OMIP2 and CMIP6 models indicates that the attribution of the interdecadal AMOC variation is not exclusive and to some degree model dependent, as summarized in Supplementary Note 2. The supporting figures and related discussion are now added in the supplementary information (Supplementary Note 2).

Supplementary Figure 4. Interdecadal time series of the externally forced AMOC and its rate of change at 26.5°N in low and medium resolution HadGEM3-GC31. (a) Interdecadal time series of the externally forced AMOC and (b) its rate of change at 26.5°N derived from HadGEM3-GC31-MM (red), HadGEM3-GC31-LL (sky blue) and the difference between the two (purple). The error bars in (a) indicate standard deviation from the ensemble-mean.

Supplementary Figure 5. Interdecadal time series of the AMOC and its rate of change at 26.5°N based on low and medium resolution HadGEM3-GC31. (a,b) Interdecadal time series of the AMOC and (b,d) its rate of change at 26.5°N based on (a,b) HadGEM3-GC31-LL and (c,d) HadGEM3-GC31-MM. The error bars in (a,c) indicate standard deviation from the ensemble-mean.

>

Jones, G. S. et al. The HadGEM3-GC3.1 contribution to the CMIP6 detection and attribution model intercomparison project. *J. Adv. Model. Earth Sys.* 16, e2023MS004135 (2024).

>

Note: There are also two high-resolution (~10km) coupled models forced under the CMIP5 RCP8.5 scenario (HadGEM3-GC31-HH and CESM1-3; Roberts et al., 2019). However, those high-resolution model runs are available only for a single ensemble member, which makes it difficult to separate the forced signal from natural variability. So, we could not use those high-resolution model runs. Among the coupled models participating in HighResMIP, we found one model (CNRM-CM6-1) with multi ensemble members (3 members in this case) available in both low (~100km) and medium (~25km) resolutions. However, HighResMIP models were forced under the CMIP5 RCP8.5 scenario. Thus, it is challenging to compare these results with the CMIP6 models (under the SSP-370 and -585 scenarios) used in our study. So, we decided not to use these HighResMIP model runs.

3. The authors argue that subtropical AMOC is influenced by time-integrated NAO variability (in contrast to subpolar AMOC), which is physically possible. So why not show the correlation between AMOC and time-integrated NAO instead of NAO index?

>

As shown in Supplementary Figure 8, both the NAO and time integrated NAO in CESM2 are positively correlated with the AMOC at 26.5N. However, in both cases the correlation is statistically insignificant. This result suggests that neither the NAO nor time-integrated NAO is the primary forcing that determines the natural variability of the AMOC in CESM2. This is consistent with Xu et al. (2019) where they concluded “..... unlike the CORE-II simulations, the CMIP5 simulations do not exhibit a robust NAO-AMOC Linkage”. Although it is not entirely clear why NAO’s impact on interdecadal AMOC variations is much smaller in CESM2 than in OMIP2, Kim et al. (2018) pointed out that the NAO simulated in fully coupled models tends to display weaker-than-observed interdecadal variations, and thus its influence on interdecadal AMOC variations may be underestimated. Kim et al. (2018) is briefly discussed in the revised manuscript.

>

Kim, W. M., Yeager, S., Chang, P., & Danabasoglu, G. Low-frequency North Atlantic climate variability in the Community Earth System Model Large Ensemble. *J. Climate* 31, 787–813 (2018).

Supplementary Figure 8. Lead-lag correlations of the NAO, the time-integrated NAO and the residual AMOC at 26.5°N. Lead-lag correlations (a) between the NAO and the residual AMOC at 26.5°N, and (b) between the time-integrated NAO and the residual AMOC at 26.5°N derived from CESM2. The green lines in (a,b) indicate the ensemble-averaged correlation values. The orange lines in (a,b) indicate the ensemble spread (i.e., standard deviation). Thick horizontal lines in (a,b) indicate the 95% confidence intervals (based on a Student-t test) and zero correlations. The degree of freedom for the Student-t test is determined by the number of decades since the NAO, time-integrated NAO and AMOC indices are all smoothed by performing a 10-year running-average prior to the correlation analysis to focus on interdecadal time scale. The units for the residual AMOC are Sv ($10^6 m^3 sec^{-1}$).

Minor Comments:

4. Why the authors adjust annual variation of the AMOC while the focus of the study is interdecadal variation?

>

In the original manuscript, we used both “annual AMOC” and “annual mean AMOC”. To clarify, they are now all consolidated into “mean AMOC” or “annual mean AMOC:” if necessary.

5. In several places of the text, the difference between total and externally force AMOC is referred to as residual AMOC. Residual AMOC is often used for describing total AMOC which is the sum of Eulerian AMOC and eddy-driven AMOC. I therefore suggest to rephrase it as residual component of the AMOC or the residual AMOC component.

>

Done

6. Please provide the horizontal and vertical resolution for models in supplementary tables or Methods.

>

The horizontal and vertical resolutions for all models used are now listed in Supplementary Tables 1-3.